# Diff-Instruct++: Training One-step Text-to-image Generator Model to Align with Human Preferences

**Weijian Luo**[*]                                                                                   *luoweijian@stu.pku.edu.cn*
*Peking University*

**Reviewed on OpenReview:** *https://openreview.net/forum?id=SeGNvJJjbs*

## Abstract

One-step text-to-image generator models offer advantages such as swift inference efficiency, flexible architectures, and state-of-the-art generation performance. In this paper, we study the problem of aligning one-step generator models with human preferences for the first time. Inspired by the success of reinforcement learning using human feedback (RLHF), we formulate the alignment problem as maximizing expected human reward functions while adding an Integral Kullback-Leibler divergence term to prevent the generator from diverging. By overcoming technical challenges, we introduce Diff-Instruct++ (DI++), the first, fast-converging and image data-free human preference alignment method for one-step text-to-image generators. We also introduce novel theoretical insights, showing that using CFG for diffusion distillation is secretly doing RLHF with DI++. Such an interesting finding brings understanding and potential contributions to future research involving CFG. In the experiment sections, we align both UNet-based and DiT-based one-step generators using DI++, which use the Stable Diffusion 1.5 and the PixelArt-$\alpha$ as the reference diffusion processes. The resulting DiT-based one-step text-to-image model achieves a strong Aesthetic Score of 6.19 and an Image Reward of 1.24 on the COCO validation prompt dataset. It also achieves a leading Human preference Score (HPSv2.0) of 28.48, outperforming other open-sourced models such as Stable Diffusion XL, DMD2, SD-Turbo, as well as PixelArt-$\alpha$. Both theoretical contributions and empirical evidence indicate that DI++ is a strong human-preference alignment approach for one-step text-to-image models. The homepage of the paper is: `https://github.com/pkulwj1994/diff_instruct_pp`.

## 1 Introductions

In recent years, deep generative models have achieved remarkable successes across various data generation and manipulation applications (Karras et al., 2020; 2022; Nichol & Dhariwal, 2021; Oord et al., 2016; Ho et al., 2022; Poole et al., 2022; Hoogeboom et al., 2022; Kim et al., 2022; Tashiro et al., 2021; Kingma & Dhariwal, 2018; Chen et al., 2019; Meng et al., 2021; Couairon et al., 2022; Zhang et al., 2023a; Luo & Zhang, 2024; Xue et al., 2023; Luo et al., 2023c; Zhang et al., 2023b; Feng et al., 2023; Deng et al., 2024; Luo et al., 2024c; Geng et al., 2024; Wang et al., 2024; Pokle et al., 2022). These models have notably excelled in producing high-resolution, text-conditional models such as images (Rombach et al., 2022; Saharia et al., 2022; Ramesh et al., 2022; 2021; Luo et al., 2024b) and other modalities with different applications(Brooks et al., 2024; Kondratyuk et al., 2023; Evans et al., 2024; Luo et al., 2024c), pushing the boundaries of Artificial Intelligence Generated Content.

Among the spectrum of deep generative models, one-step generators have emerged as a particularly efficient and possibly best performing (Zheng & Yang, 2024; Kim et al., 2024; Kang et al., 2023; Sauer et al., 2023a) generative model. Briefly speaking, a one-step generator uses a neural network to directly transport some latent variable to generate an output sample. Recently, there have been many fruitful successes in training one-step generator models by distilling from pre-trained diffusion models (aka, Diffusion Distillation (Luo,

---

*Alternative email: pkulwj1994@icloud.com.

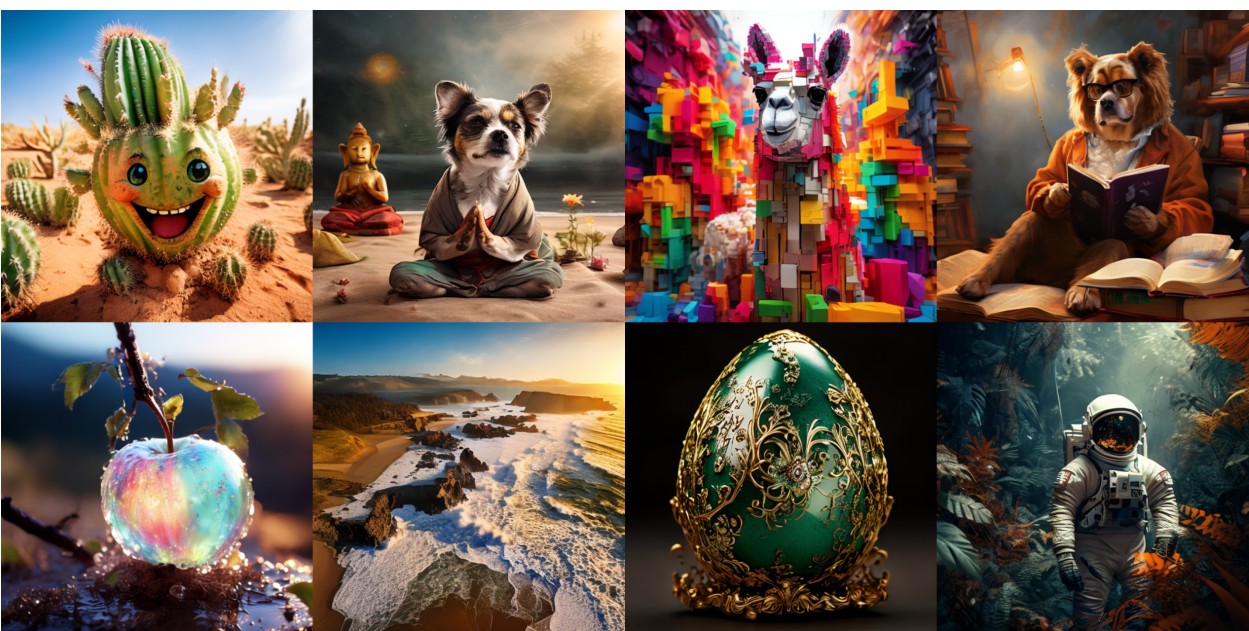

Figure 1: Images generated by a one-step text-to-image generator that has been aligned with human preferences using Diff-Instruct++. We put the prompt in Appendix D.1.

2023)) in domains of image generations (Salimans & Ho, 2022; Luo et al., 2024a; Geng et al., 2023; Song et al., 2023; Kim et al., 2023; Song & Dhariwal, 2023; Zhou et al., 2024b), text-to-image synthesis (Meng et al., 2022; Gu et al., 2023; Nguyen & Tran, 2023; Luo et al., 2024b; 2023a; Song et al., 2024; Liu et al., 2024b; Yin et al., 2023), data manipulation (Parmar et al., 2024), etc.

However, current one-step text-to-image generators face several limitations, including insufficient adherence to user prompts, suboptimal aesthetic quality, and even the generation of toxic content. These issues arise because the generator models are not aligned with human preferences. In this paper, we study the problem of aligning one-step generator models with human preferences for the first time. We achieve substantial progress by training generator models to maximize human preference reward. Inspired by the success of reinforcement learning using human feedback (RLHF) (Ouyang et al., 2022) in aligning large language models, we formulate the alignment problem as a maximization of the expected human reward function with an additional regularization term to some reference diffusion model. By addressing technical challenges, we obtain effective loss functions and formally introduce Diff-Instruct++, an image data-free method to train one-step text-to-image generators to follow human preferences.

In the experiment part, we demonstrate the strong compatibility of Diff-Instruct++ on both the UNet-based diffusion model and one-step generator such as Stable Diffusion 1.5, and the DiT-based (Peebles & Xie, 2022) diffusion and generator such as PixelArt-$\alpha$ (Chen et al., 2023). We first pre-train a one-step generator model using Diff-Instruct (Luo et al., 2024a), initialized with Stable Diffusion 1.5 and PixelArt-$\alpha$. We name this model an unaligned base generator model (base model for short). Next, we align the base model using DI++ with an off-the-shelf Image Reward (Xu et al., 2023a) model using different configurations, resulting in various human-preferred one-step text-to-image models.

The alignment process with DI++ significantly improves the generation quality of the one-step generator model with minimum computational cost. To evaluate our models from different perspectives, we conduct both qualitative and quantitative evaluations of aligned models with different alignment settings. In the quantitative evaluation, we evaluate the model with several commonly used quality metrics such as Huma-preference Score (HPSv2.0)(Wu et al., 2023), the Aesthetic score (Schuhmann, 2022), the Image Reward (Xu et al., 2023a), and the PickScore (Kirstain et al., 2023) and the CLIP score on the same 1k prompts from MSCOCO-2017 validation datasets. Our main findings are: 1) the best DiT-based one-step text-to-image model achieves a leading HPSv2.0 score of **28.48**; 2) The aligned models outperform the unaligned ones

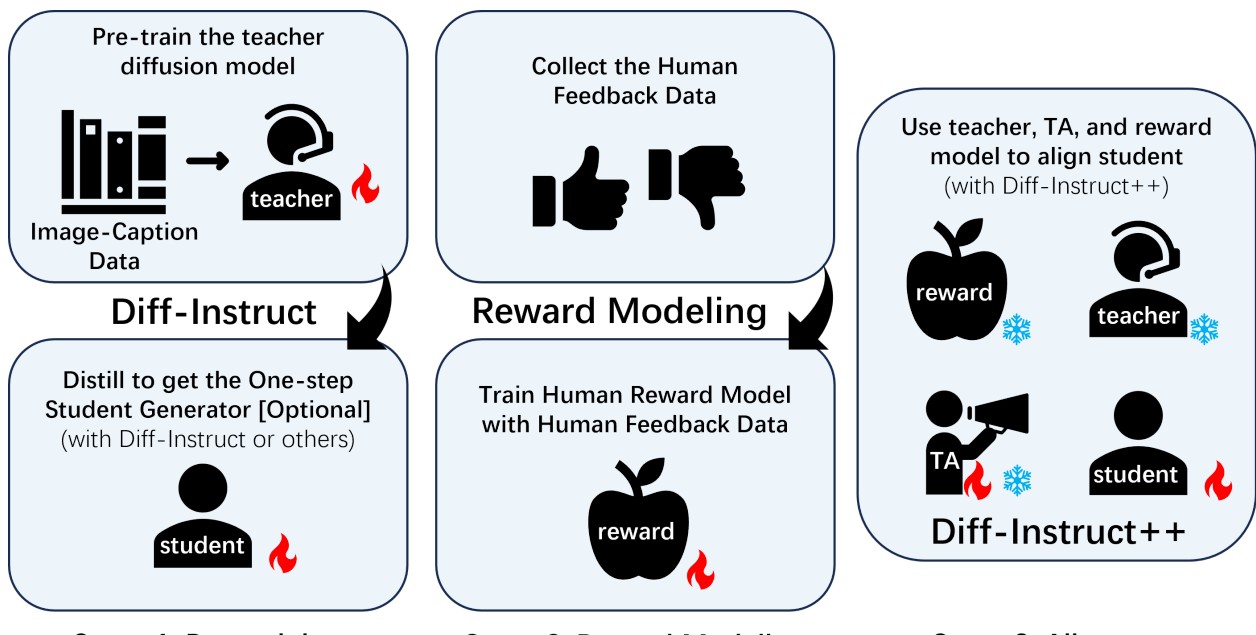

Figure 2: A demonstration of three stages for training a one-step text-to-image generator model that is aligned with human preference. The pre-training stage (**the leftmost column**) pre-trains the reference diffusion model as well as the one-step generator. The reward modeling stage (**the middle column**) trains the reward model using human preference data. The alignment stage (**the rightmost column**) uses a pre-trained reference diffusion model, the reward model, and a TA diffusion model to align the one-step generator with human preference.

with significant margins in terms of human preference metrics; 3) The alignment cost is cheap, and the convergence is fast; 4) The aligned model still makes simple mistakes.

We will discuss these findings in Section 5.3 in detail. We also establish the theoretical connections of DI++ with diffusion distillation methods, as well as the classifier-free guidance (Ho & Salimans, 2022). In Theorem 3.4 in Section 3.1, we show an interesting finding that *the well-known classifier-free guidance is secretly doing RLHF according to an implicitly-defined reward function.* Therefore, diffusion distillation with CFG can be unified within DI++. These theoretical findings not only help understand the behavior of classifier-free guidance but also bring new tools for human preference alignment for text-to-image models.

In Section 5.4, we discuss the potential shortcomings of DI++, showing that even aligned with human reward, the one-step model still makes simple mistakes sometimes. Imperfect human reward and insufficient hyperparameter tunings possibly cause this issue.

## 2 Preliminary

**Diffusion Models.** In this section, we introduce preliminary knowledge and notations about diffusion models. Assume we observe data from the underlying distribution $q_d(\boldsymbol{x})$. The goal of generative modeling is to train models to generate new samples $\boldsymbol{x} \sim q_d(\boldsymbol{x})$. Under mild conditions, the forward diffusion process of a diffusion model can transform any initial distribution $q_0 = q_d$ towards some simple noise distribution,

$$\mathrm{d}\boldsymbol{x}_t = \boldsymbol{F}(\boldsymbol{x}_t, t)\mathrm{d}t + G(t)\mathrm{d}\boldsymbol{w}_t, \tag{2.1}$$

where $\boldsymbol{F}$ is a pre-defined vector-valued drift function, $G(t)$ is a pre-defined scalar-value diffusion coefficient, and $\boldsymbol{w}_t$ denotes an independent Wiener process. A continuous-indexed score network $\boldsymbol{s}_\varphi(\boldsymbol{x}, t)$ is employed to approximate marginal score functions of the forward diffusion process (2.1). The learning of score networks

is achieved by minimizing a weighted denoising score matching objective (Vincent, 2011; Song et al., 2020),

$$\mathcal{L}_{DSM}(\varphi) = \int_{t=0}^{T} \lambda(t) \mathbb{E}_{\substack{\boldsymbol{x}_0 \sim q_0, \\ \boldsymbol{x}_t | \boldsymbol{x}_0 \sim q_t(\boldsymbol{x}_t | \boldsymbol{x}_0)}} \| \boldsymbol{s}_{\varphi}(\boldsymbol{x}_t, t) - \nabla_{\boldsymbol{x}_t} \log q_t(\boldsymbol{x}_t | \boldsymbol{x}_0) \|_2^2 \mathrm{d}t. \tag{2.2}$$

Here the weighting function $\lambda(t)$ controls the importance of the learning at different time levels and $q_t(\boldsymbol{x}_t | \boldsymbol{x}_0)$ denotes the conditional transition of the forward diffusion (2.1). After training, the score network $\boldsymbol{s}_{\varphi}(\boldsymbol{x}_t, t) \approx \nabla_{\boldsymbol{x}_t} \log q_t(\boldsymbol{x}_t)$ is a good approximation of the marginal score function of the diffused data distribution.

**Reinforcement Learning using Human Feedback.** Reinforcement learning using human feedback (Christiano et al., 2017; Ouyang et al., 2022) (RLHF) is originally proposed to incorporate human feedback knowledge to improve large language models (LLMs). Let $p_\theta(\boldsymbol{x}|\boldsymbol{c})$ be a large language model's output distribution, where $\boldsymbol{c}$ is the input prompt that is randomly sampled from a prompt dataset $\mathcal{C}$, and the $\boldsymbol{x}$ is the generated responses. Let $r(\boldsymbol{x}, \boldsymbol{c})$ be a scalar reward model that probably has been trained with human feedback data and thus can measure the human preference on an image-prompt pair $(\boldsymbol{x}, \boldsymbol{c})$. Let $p_{ref}(\boldsymbol{x}|\boldsymbol{c})$ be some reference LLM model. The RLHF method trains the LLM to maximize the human reward with a Kullback-Leibler divergence regularization, which is equivalent to minimizing:

$$\mathcal{L}(\theta) = \mathbb{E}_{\substack{\boldsymbol{c} \sim \mathcal{C}, \\ \boldsymbol{x} \sim p_\theta(\boldsymbol{x}|\boldsymbol{c})}} \big[ -r(\boldsymbol{x}, \boldsymbol{c}) \big] + \beta \mathcal{D}_{KL}(p_\theta(\boldsymbol{x}|\boldsymbol{c}), p_{ref}(\boldsymbol{x}|\boldsymbol{c})) \tag{2.3}$$

The KL divergence regularization term lets the model be close to the reference model thus preventing it from diverging, while the reward term encourages the model to generate outputs with high rewards. After the RLHF, the model will be aligned with human preference.

**One-step Text-to-image Generator Model.** A one-step (Goodfellow et al., 2014; Luo et al., 2024a; Yin et al., 2023; Zhou et al., 2024b; Luo et al., 2024b) text-to-image generator model is a neural network $g_\theta(\cdot|\cdot)$, that can turn an input latent variable $\boldsymbol{z} \sim p_z(\boldsymbol{z})$ and an input prompt $\boldsymbol{c}$ to a generate image $\boldsymbol{x}$ (or some latent vector before decoding as the case of latent diffusion models (Rombach et al., 2022)) with a single neural network forward inference: $\boldsymbol{x} = g_\theta(\boldsymbol{z}|\boldsymbol{c})$. Compared with diffusion models, the one-step generator model has advantages such as fast inference speed and flexible neural architectures.

## 3 Human-preference Alignment of One-step Text-to-image Models

In this section, we introduce how to align one-step text-to-image generator models and human preferences using Diff-Instruct++. In Section 3.1, we introduce the formulation of the alignment problem and then identify the alignment objective. After that, we address technical challenges and propose the Theorem 3.1 and Theorem 3.2, which set the theoretical foundation of DI++ through the lens of reward maximization. Interestingly, we also find that the diffusion distillation using classifier-free guidance is secretly doing RLHF with DI++. Based on the theoretical arguments in Section 3.1, we formally introduce the practical algorithm of DI++ in Section 3.3, and give an intuitive understanding of the algorithm as an education process that includes a teacher diffusion model, a teaching assistant diffusion model, and a student one-step generation.

### 3.1 The Alignment Objective

**The Problem Formulation.** We consider the text-to-image generation task. Other conditional generation applications share a similar spirit. Assume $\boldsymbol{x}$ is an image and $\boldsymbol{c}$ is a text prompt that is sampled from some prompt dataset $\mathcal{C}$. The basic setting is that we have a one-step generator $g_\theta(\cdot|\cdot)$, which can transport a prior latent vector $\boldsymbol{z} \sim p_z$ to generate an image based on input text prompt $\boldsymbol{c}$: $\boldsymbol{x} = g_\theta(\boldsymbol{z}|\boldsymbol{c})$. We use the notation $p_\theta(\boldsymbol{x}|\boldsymbol{c})$ to denote the distribution induced by the generator. We also have a reward model $r(\boldsymbol{x}, \boldsymbol{c})$ which represents the human preference on a given image-prompt pair $(\boldsymbol{x}, \boldsymbol{c})$.

Inspired by the success of reinforcement learning using human feedback (Ouyang et al., 2022) in fine-tuning large language models such as ChatGPT (Achiam et al., 2023), we first set our alignment objective to maximize the expected human reward function with an additional Kullback-Leibler divergence regularization

w.r.t some reference distribution $p_{ref}(\cdot|\boldsymbol{c})$, which is equivalent to minimize the following objective:

$$
\begin{aligned}
\mathcal{L}(\theta) &= \mathbb{E}_{\substack{\boldsymbol{c}\sim\mathcal{C},\\ \boldsymbol{x}\sim p_\theta(\boldsymbol{x}|\boldsymbol{c})}} \big[ -r(\boldsymbol{x},\boldsymbol{c}) \big] + \beta \mathcal{D}_{KL}(p_\theta(\boldsymbol{x}|\boldsymbol{c}), p_{ref}(\boldsymbol{x}|\boldsymbol{c})) \\
&= \mathbb{E}_{\substack{\boldsymbol{c}\sim\mathcal{C},\boldsymbol{z}\sim p_z,\\ \boldsymbol{x}=g_\theta(\boldsymbol{z}|\boldsymbol{c})}} \big[ -r(\boldsymbol{x},\boldsymbol{c}) \big] + \beta \mathcal{D}_{KL}(p_\theta(\boldsymbol{x}|\boldsymbol{c}), p_{ref}(\boldsymbol{x}|\boldsymbol{c}))
\end{aligned}
\tag{3.1}
$$

The KL divergence regularization to the reference distribution $p_{ref}(\cdot)$ guarantees the generator distribution $p_\theta(\cdot)$ to stay similar to $p_{ref}(\cdot)$ in order to prevent it from diverging.

Though objective (3.1) is appealing, for the one-step model, we do not know the explicit form of distribution $p_\theta(\boldsymbol{x}|\boldsymbol{c})$. Inspired by Diff-Instruct(Luo et al., 2024a) and other distribution matching-based diffusion distillation approaches, we can effectively estimate the $\theta$-gradient of objective (3.1) with equation (3.2) in Theorem 3.1.

**Theorem 3.1.** The $\theta$ gradient of the objective (3.1) is

$$
\mathrm{Grad}(\theta) = \mathbb{E}_{\substack{\boldsymbol{c}\sim\mathcal{C},\boldsymbol{z}\sim p_z,\\ \boldsymbol{x}=g_\theta(\boldsymbol{z}|\boldsymbol{c})}} \left\{ -\nabla_{\boldsymbol{x}}r(\boldsymbol{x},\boldsymbol{c}) + \beta\big[\nabla_{\boldsymbol{x}}\log p_\theta(\boldsymbol{x}|\boldsymbol{c}) - \nabla_{\boldsymbol{x}}\log p_{ref}(\boldsymbol{x}|\boldsymbol{c})\big] \right\} \frac{\partial \boldsymbol{x}}{\partial \theta}
\tag{3.2}
$$

We will give the proof in Appendix B.1. For gradient formula (3.2), we can see that the $\boldsymbol{x}$ gradient of $r(\boldsymbol{x},\boldsymbol{c})$ is easy to obtain. If we can approximate the score function of both generator and reference distribution, i.e. $\nabla_{\boldsymbol{x}}\log p_\theta(\boldsymbol{x}|\boldsymbol{c})$ and $\nabla_{\boldsymbol{x}}\log p_{ref}(\boldsymbol{x}|\boldsymbol{c})$, we can directly compute the $\theta$ gradient and use gradient descent algorithms to update the parameters $\theta$. However, since the generator distribution is defined directly in image space, where the distributions are assumed to lie on some low dimensional manifold (Song & Ermon, 2019). Therefore, *approximating the score function and minimizing KL divergence is difficult in practice.*

**Diffusion Models are Reference Processes.** Instead of minimizing the negative reward with KL regularization in (3.1), we turn to (3.3) by generalizing the KL divergence regularization to the Integral Kullback-Leibler divergence proposed in Luo et al. (2024a) w.r.t to some reference diffusion process $p_{ref}(\boldsymbol{x}_t|t,\boldsymbol{c})$. This novel change of regularization divergence distinguishes our approach from RLHF methods for large language model alignments. Besides, such a change from KL divergence to IKL divergence makes it possible to *use pre-trained diffusion models as reference processes*, as we will show in the following paragraphs.

Let $\boldsymbol{x}_t$ be noisy data that is diffused by the forward diffusion (2.1) starting from $\boldsymbol{x}_0$. We use $p_{ref}(\boldsymbol{x}_t|t,\boldsymbol{c})$ and $\boldsymbol{s}_{ref}(\boldsymbol{x}_t|t,\boldsymbol{c})$ to denote the densities and score functions of the reference diffusion process (the score functions can be replaced with pre-trained off-the-shelf diffusion models). Let $p_\theta(\boldsymbol{x}|t,\boldsymbol{c})$ and $\boldsymbol{s}_\theta(\boldsymbol{x}_t|t,\boldsymbol{c})$ be the marginal distribution and score functions of the generator output after forward diffusion process (2.1). We propose to minimize the negative reward function with an Integral KL divergence regularization:

$$
\mathcal{L}(\theta) = \mathbb{E}_{\substack{\boldsymbol{c},\boldsymbol{z}\sim p_z,\boldsymbol{x}_0=g_\theta(\boldsymbol{z}|\boldsymbol{c})\\ \boldsymbol{x}_t|\boldsymbol{x}_0\sim p(\boldsymbol{x}_t|\boldsymbol{x}_0)}} \big[ -r(\boldsymbol{x}_0,\boldsymbol{c}) \big] + \beta \int_{t=0}^{T} w(t)\mathcal{D}_{KL}(p_\theta(\boldsymbol{x}_t|t,\boldsymbol{c}), p_{ref}(\boldsymbol{x}_t|t,\boldsymbol{c}))\mathrm{d}t
\tag{3.3}
$$

Different from vanilla RLHF objective (3.1), the objective with IKL regularization (3.3) assigned a regularization between generator's noisy distributions and a reference diffusion process $p_{ref}(\cdot|t,\boldsymbol{c})$. Following similar spirits of Theorem 3.1, we have a gradient formula in Theorem 3.2 which takes the IKL divergence into account.

**Theorem 3.2.** The $\theta$ gradient of the objective (3.3) is

$$
\mathrm{Grad}(\theta) = \mathbb{E}_{\substack{\boldsymbol{c},t,\boldsymbol{z}\sim p_z,\boldsymbol{x}_0=g_\theta(\boldsymbol{z}|\boldsymbol{c})\\ \boldsymbol{x}_t|\boldsymbol{x}_0\sim p(\boldsymbol{x}_t|\boldsymbol{x}_0)}} \left\{ -\nabla_{\boldsymbol{x}_0}r(\boldsymbol{x}_0,\boldsymbol{c}) + \beta w(t)\big[\boldsymbol{s}_\theta(\boldsymbol{x}_t|t,\boldsymbol{c}) - \boldsymbol{s}_{ref}(\boldsymbol{x}_t|t,\boldsymbol{c})\big]\frac{\partial \boldsymbol{x}_t}{\partial \theta} \right\}
\tag{3.4}
$$

We will give the proof in Appendix B.2. We can clearly see that the reference score functions can be replaced with off-the-shelf text-to-image diffusion models.

**Remark 3.3.** If we view the generator as a *student*, the reference diffusion model acts like a *teacher*. We can use another diffusion model $\boldsymbol{s}_\psi(\cdot)$ to act like a *teaching assistant (TA)*, which is initialized from the teacher and fine-tuned using student-generated data to approximate the student generator score functions,

i.e. $\boldsymbol{s}_\psi(\boldsymbol{x}_t|t,\boldsymbol{c}) \approx \boldsymbol{s}_\theta(\boldsymbol{x}_t|t,\boldsymbol{c})$. The reward function $r(\cdot,\cdot)$ can be viewed as the student's *personal preference*. With this, we can readily estimate the gradient (3.4) and update the generator model to maximize students' interests (aka, to maximize the reward) while still referring to the teacher's and the TA's advice. From this perspective, the Diff-Instruct++ algorithm is similar to an education procedure that encourages the student to maximize personal interests, while still taking advice from teachers and teaching assistants. Since the use of IKL divergence for regularization in RLHF is inspired by Diff-Instruct (Luo et al., 2024a), we name our alignment approach the Diff-Instruct++.

Though minimizing objective (3.3) is solid in theory, it would be beneficial to properly understand and use a classifier-free guidance mechanism in the alignment process. In the following paragraphs, we give an interesting theoretical finding, showing that classifier-free guidance is secretly doing RLHF with an implicitly-defined reward function. With this, we can properly incorporate both the human reward and the CFG reward for alignment with Diff-Instruct++.

## 3.2 Classifier-free Guidance is Secretly Doing RLHF.

In previous sections, we have shown in theory that with available reward models, we can readily do RLHF for one-step generators. However, in this part, we additionally find that the classifier-free guidance is secretly doing RLHF, and therefore we will show that using CFG for reference diffusion models when distilling them using Diff-Instruct is secretly doing Diff-Instruct++ with an implicitly defined reward.

The classifier-free guidance (Ho & Salimans, 2022) (CFG) uses a score function with a form

$$\widetilde{\boldsymbol{s}}_{ref}(\boldsymbol{x}_t,t|\boldsymbol{c}) := \boldsymbol{s}_{ref}(\boldsymbol{x}_t,t|\boldsymbol{\varnothing}) + \omega\big\{\boldsymbol{s}_{ref}(\boldsymbol{x}_t,t|\boldsymbol{c}) - \boldsymbol{s}_{ref}(\boldsymbol{x}_t,t|\boldsymbol{\varnothing})\big\}$$

to replace the original conditions score function $\boldsymbol{s}_{ref}(\boldsymbol{x}_t,t|\boldsymbol{c})$. This empirically leads to better sampling quality for diffusion models. In this part, we show how diffusion distillation using Diff-Instruct with CFG is secretly doing RLHF with DI++. If we consider a reward function as:

$$r(\boldsymbol{x}_0,\boldsymbol{c}) = \int_{t=0}^{T} w(t)\log\frac{p_{ref}(\boldsymbol{x}_t|t,\boldsymbol{c})}{p_{ref}(\boldsymbol{x}_t|t)}\mathrm{d}t. \tag{3.5}$$

This reward function will put a higher reward to those samples that have higher conditional probability than unconditional probability. Therefore, it can encourage samples with high conditional probability. We show that the gradient formula (3.4) in Theorem 3.2 will have an explicit solution:

**Theorem 3.4.** Under mild conditions, if we set an implicit reward function as (3.5), the gradient formula (3.4) in Theorem 3.2 with have an explicit expression:

$$\text{Grad}(\theta) = \mathbb{E}_{\substack{\boldsymbol{c},t,\boldsymbol{z}\sim p_z, \boldsymbol{x}_0 = g_\theta(\boldsymbol{z}|\boldsymbol{c}) \\ \boldsymbol{x}_t|\boldsymbol{x}_0\sim p(\boldsymbol{x}_t|\boldsymbol{x}_0)}} \beta w(t)\bigg\{\boldsymbol{s}_\theta(\boldsymbol{x}_t|t,\boldsymbol{c}) - \widetilde{\boldsymbol{s}}_{ref}^{\beta}(\boldsymbol{x}_t|t,\boldsymbol{c})\bigg\}\frac{\partial\boldsymbol{x}_t}{\partial\theta} \tag{3.6}$$

$$\widetilde{\boldsymbol{s}}_{ref}^{\beta}(\boldsymbol{x}_t|t,\boldsymbol{c}) = \boldsymbol{s}_{ref}(\boldsymbol{x}_t|t) + (1+\frac{1}{\beta})\big[\boldsymbol{s}_{ref}(\boldsymbol{x}_t|t,\boldsymbol{c}) - \boldsymbol{s}_{ref}(\boldsymbol{x}_t|t)\big]$$

We will give the proof in Appendix B.4. This gradient formula (3.6) recovers the case that uses the CFG for diffusion distillation using the Diff-Instruct algorithm to train a one-step generator. The parameter $(1+\frac{1}{\beta})$ is the so-called classifier-free guidance scale. In our Algorithm 1 and 2, we use the coefficient $\alpha_{cfg}$ to represent the CFG scale. In the following section, we formally introduce the practical algorithm of Diff-Instruct++.

**Remark 3.5.** Theorem 3.4 has revealed a new perspective that understands classifier-free guidance as a training-free and inference-time RLHF. This helps to understand why humans prefer samples generated by using CFG. Besides, Theorem 3.4 also shows that using Diff-Instruct with CFG to distill text-to-image diffusion models is secretly doing DI++. Therefore we can use both CFG and the human reward to strengthen the one-step generator models.

## 3.3 A Practical Algorithm

Though the gradient formula (3.2) gives an intuitive way to compute the parameter gradient to update the generator, it would be better to have an easy-to-implement pseudo loss function instead of the gradient

---

**Algorithm 1:** Diff-Instruct++ for aligning generator model with human feedback reward.

---

**Input:** prompt dataset $\mathcal{C}$, generator $g_\theta(\boldsymbol{x}_0|\boldsymbol{z}, \boldsymbol{c})$, prior distribution $p_z$, reward model $r(\boldsymbol{x}, \boldsymbol{c})$, reward scale $\alpha_{rew}$, CFG scale $\alpha_{cfg}$, reference diffusion model $\boldsymbol{s}_{ref}(\boldsymbol{x}_t|c, \boldsymbol{c})$, TA diffusion $\boldsymbol{s}_\psi(\boldsymbol{x}_t|t, \boldsymbol{c})$, forward diffusion $p(\boldsymbol{x}_t|\boldsymbol{x}_0)$ (2.1), TA diffusion updates rounds $K_{TA}$, time distribution $\pi(t)$, diffusion model weighting $\lambda(t)$, generator IKL loss weighting $w(t)$.

**while** *not converge* **do**

    fix $\theta$, update $\psi$ for $K_{TA}$ rounds by minimizing

$$\mathcal{L}(\psi) = \mathbb{E}_{\substack{\boldsymbol{c}\sim\mathcal{C}, \boldsymbol{z}\sim p_z, t\sim\pi(t) \\ \boldsymbol{x}_0 = g_\theta(\boldsymbol{z}|\boldsymbol{c}), \boldsymbol{x}_t|\boldsymbol{x}_0\sim p_t(\boldsymbol{x}_t|\boldsymbol{x}_0)}} \lambda(t)\|\boldsymbol{s}_\psi(\boldsymbol{x}_t|t, \boldsymbol{c}) - \nabla_{\boldsymbol{x}_t}\log p_t(\boldsymbol{x}_t|\boldsymbol{x}_0)\|_2^2 \mathrm{d}t.$$

    update $\theta$ using StaD with the gradient

$$\mathrm{Grad}(\theta) = \mathbb{E}_{\substack{\boldsymbol{c}\sim\mathcal{C}, \boldsymbol{z}\sim p_z, \\ \boldsymbol{x}_0 = g_\theta(\boldsymbol{z}, \boldsymbol{c})}} \left[-\alpha_{rew}\nabla_{\boldsymbol{x}_0} r(\boldsymbol{x}_0, \boldsymbol{c})\right] \tag{3.7}$$

$$+ \int_{t=0}^T w(t)\mathbb{E}_{\substack{\boldsymbol{c}\sim\mathcal{C}, \boldsymbol{z}\sim p_z, t\sim\pi(t) \\ \boldsymbol{x}_0 = g_\theta(\boldsymbol{z}, \boldsymbol{c}), \boldsymbol{x}_t|\boldsymbol{x}_0\sim p_t(\boldsymbol{x}_t|\boldsymbol{x}_0)}} \left\{\boldsymbol{s}_\psi(\boldsymbol{x}_t|t, \boldsymbol{c}) - \widetilde{\boldsymbol{s}}_{ref}(\boldsymbol{x}_t|t, \boldsymbol{c})\right\}\frac{\partial \boldsymbol{x}_t}{\partial \theta}\mathrm{d}t. \tag{3.8}$$

$$\widetilde{\boldsymbol{s}}_{ref}(\boldsymbol{x}_t|t, \boldsymbol{c}) = \boldsymbol{s}_{ref}(\boldsymbol{x}_t|t, \boldsymbol{\varnothing}) + \alpha_{cfg}\left[\boldsymbol{s}_{ref}(\boldsymbol{x}_t|t, \boldsymbol{c}) - \boldsymbol{s}_{ref}(\boldsymbol{x}_t|t, \boldsymbol{\varnothing})\right] \tag{3.9}$$

**end**

**return** $\theta, \psi$.

---

estimations for executing the algorithm. To address such an issue, we present a pseudo loss function defined as formula (B.13), which we show has the same gradient as (3.4) in Appendix.

With the pseudo loss function (B.13), we formally introduce the DI++ algorithm which is presented in Algorithm 1 (and a more executable version in Algorithm 2). As in Algorithm 1, the overall algorithms consist of two alternative updating steps. The first step update $\psi$ of the TA diffusion model by fine-tuning it with student-generated data. Therefore the TA diffusion $\boldsymbol{s}_\psi(\boldsymbol{x}_t|t, \boldsymbol{c})$ can approximate the score function of student generator distribution. This step means that the TA needs to communicate with the student to know the student's status. The second step estimates the parameter gradient of equation (3.2) and uses this parameter gradient for gradient descent optimization algorithms such as Adam (Kingma & Ba, 2014) to update the generator parameter $\theta$. This step means that the teacher and the TA discuss and incorporate the student's interests to instruct the student generator. Due to page limitations, we put a discussion about the meanings of the hyper-parameters in Appendix A.2.

## 4 Related Works

**RLHF for Large Language Models.** Reinforcement learning using human feedback (RLHF) has won great success in aligning large language models (LLMs). Ouyang et al. (2022) formulates the LLM alignment problem as maximization of human reward with a KL regularization to some reference LLM, resulting in the InstructGPT model. The Diff-Instruct++ draws inspiration from Ouyang et al. (2022). However, DI++ differs from RLHF for LLMs in several aspects: the introduction of IKL regularization, the novel gradient formula, and the overall algorithms. Many variants of RLHF for LLMs have also been intensively studied in Tian et al. (2023); Christiano et al. (2017); Rafailov et al. (2024); Ethayarajh et al. (2024), etc.

**Diffusion Distillation Through Divergence Minimization.** Diffusion distillation (Luo, 2023) is a research area that aims to reduce generation costs using teacher diffusion models. Among all existing methods, one important line is to distill a one-step generator model by minimizing certain divergences between one-step generator models and some pre-trained diffusion models. (Luo et al., 2024a) first study the diffusion distillation by minimizing the Integral KL divergence. Yin et al. (2023) generalize such a concept and add a data regression loss to distill pre-trained Stable Diffusion Models. Many other works have introduced additional techniques and improved the distillation performance (Geng et al., 2023; Kim et al., 2023; Song et al., 2023; Song & Dhariwal; Nguyen & Tran, 2023; Song et al., 2024; Yin et al., 2024;

Zhou et al., 2024a; Heek et al., 2024; Xie et al., 2024; Salimans et al., 2024). Different from IKL divergence, Zhou et al. (2024b) study the distillation through a variant of Fisher divergence. Other methods, such as Xiao et al. (2021); Xu et al. (2024), have used the generative adversarial training (Goodfellow et al., 2014) techniques in order to minimize certain divergences. The Diff-Instruct++ is motivated by Diff-Instruct. However, to the best of our knowledge, we are the first to study the problem of aligning one-step generator models with human preferences.

**Preference Alignment for Diffusion Models.** In recent years, many works have emerged trying to align diffusion models with human preferences. There are three main lines of alignment methods for diffusion models. 1) The first kind of method fine-tunes the diffusion model over a specifically curated image-prompt dataset (Dai et al., 2023; Podell et al., 2023). 2) the second line of methods tries to maximize some reward functions either through the multi-step diffusion generation output (Prabhudesai et al., 2023; Clark et al., 2023; Lee et al., 2023) or through policy gradient-based RL approaches (Fan et al., 2024; Black et al., 2023). Though this approach shares goals similar to those of DI++ and RLHF for LLMs, the problems and challenges are essentially different. Our Diff-Instruct++ is the first work to study the alignment of one-step generator models, instead of diffusion models. Besides, backpropagating through the multi-step diffusion generation output is expensive and hard to scale. 3) the third line, such as Diffusion-DPO (Wallace et al., 2024), Diffusion-KTO (Yang et al., 2024), tries to directly improve the diffusion model's human preference property with raw collected data instead of reward functions.

## 5 Experiments

In previous sections, we have established the theoretical foundations for DI++. In this section, we consider one-step text-to-image models with two kinds of neural network architectures: the UNet-based one-step text-to-image model with the well-known Stable Diffusion 1.5 (SD1.5)(Rombach et al., 2022) as the reference diffusion model, and the diffusion-transformer-based one-step model with the PixelArt-$\alpha$ as the reference diffusion. For both models, we use DI++ to align the one-step models with human preferences using ImageReward(Xu et al., 2023a).

### 5.1 The Experiment Setup

As we have shown in Figure 2, our experiment workflow consists of three modeling stages: the pre-training stage, the reward modeling stage (not necessary), and the alignment stage.

**The Pre-training Stage.** As the leftmost column of Figure 2 shows, the pre-training stage pre-trains the reference diffusion model, and a base one-step generator that is not necessarily aligned with human preference. The researcher can either train the diffusion model in-house or just use publicly available off-the-shelf diffusion models. With the pre-trained teacher diffusion model, we can pre-train our one-step generator model using diffusion distillation methods (Luo, 2023), generative adversarial training (Sauer et al., 2023a; Zheng & Yang, 2024), or a combination of them (Kim et al., 2023; Yin et al., 2024; Xu et al., 2024). Notice that in the pre-training stage, we do not necessarily use human reward to instruct the one-step model, therefore the generator can only learn to match the reference distribution, leading to decent generation results without strictly following human preference.

**The Reward Modeling Stage.** As the middle column of Figure 2 shows, in the second stage, the researcher is supposed to collect human feedback data and train a reward model that reflects human preferences for images and corresponding captions. Notice that the image and caption data for this stage can either be real image data or those images and prompts generated by users using the one-step generators in the first stage. For instance, if researchers want to enhance the generation quality of their users' commonly used prompts, they can collect the most used prompts from their users' activity and generate images using the one-step generator pre-trained in the first stage. They can send the image-prompt pair to users for feedback on their preferences. The reward modeling method is quite flexible. Researchers can either train the reward

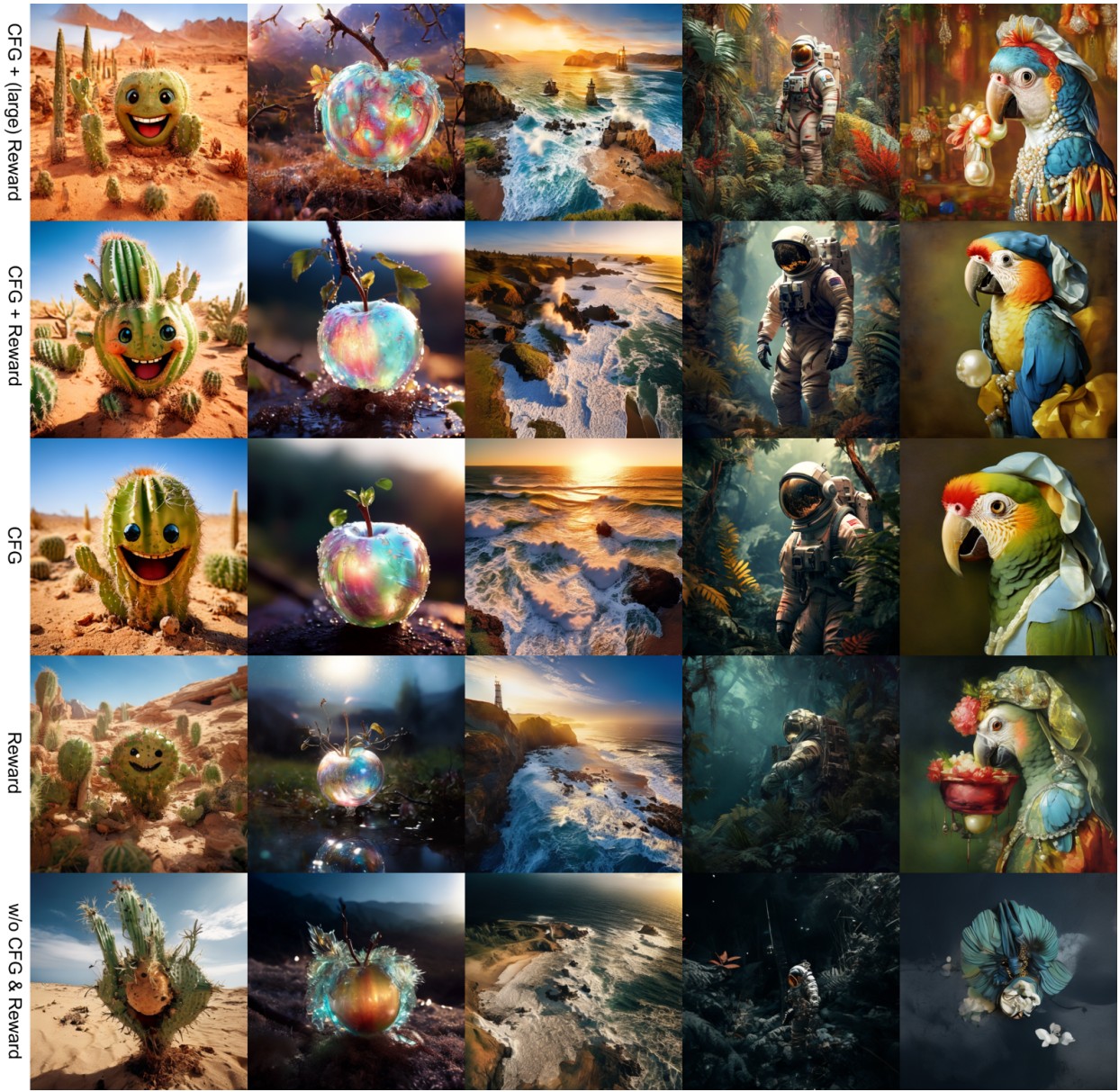

Figure 3: A qualitative comparison of one-step generator models aligned using Diff-Instruct++ with different configurations. The bottom row is the weakest setting with no human preference alignment. Upper rows are models that are aligned stronger in a progressive way. We put the prompts to generate images in Appendix D.2. The generated images are more and more aesthetic with stronger human preference alignments.

model in-house using different neural networks and data (Ouyang et al., 2022), or using off-the-shelf reward models such as Image Reward[1] (Xu et al., 2023a), or human preference scores (Wu et al., 2023).

**The Alignment Stage.** As the rightmost column of Figure 2 shows, the final stage is the alignment stage, which is the major stage that DI++ considers. In this stage, researchers can use the reference diffusion model from the first stage as a teacher, and the reward model from the second stage to align the one-step generator with the DI++ algorithm (Algorithm 1 or 2). After the alignment stage, the one-step generator can generate images that are not only realistic but also match human preferences.

---

[1] https://github.com/THUDM/ImageReward

**The Models and the Dataset.** For the SD1.5 experiment, we refer to the high-quality codebase of SiD-LSG (Zhou et al., 2024a) when constructing a 0.86B UNet-based one-step text-to-image model. For the PixelArt-$\alpha$ experiment, we use the 0.6B PixelArt-$\alpha$ model (Chen et al., 2023) of 512×512 resolution to initialize and construct an 0.6B one-step model. We put more experiment details in Appendix A.3

For the SD1.5 experiment, we use prompts from the Laion-Aesthetic dataset with an Aesthetic score higher than 6.25 (i.e., Laion-Aesthetic 625+ dataset) following the SiD-LSG. For the PixelArt-$\alpha$ experiment, we use the prompts from the SAM-LLaVA-Caption-10M dataset as our prompt dataset. The SAM-LLaVA-Caption-10M dataset contains the images collected by Kirillov et al. (2023), together with text descriptions that are captioned by LLaVA model (Liu et al., 2024a). The SAM-LLaVA-Caption-10M dataset is used for training the PixelArt-$\alpha$ model. Since the PixelArt-$\alpha$ diffusion model uses a T5-XXL, which is memory and computationally expensive. To speed up the alignment training, we pre-encode the text prompts using the T5-XXL text encoder and save the encoded embedding vectors.

## 5.2 Quantitative Evaluations with Standard Scores

In this section, we quantitatively evaluate one-step models aligned with DI++ together with other text-to-image generative models.

**Evaluation Metrics.** We evaluate five widely used human preference scores: the human preference score (HPSv2.0) (Wu et al., 2023), the ImageReward(Xu et al., 2023a), the Aesthetic Score (Schuhmann, 2022), the PickScore(Kirstain et al., 2023), and the CLIPScore(Radford et al., 2015). Among them, the HPSv2.0 has a standard prompt dataset for evaluations, therefore we follow its tradition to evaluate model scores. For the other four scores, we pick 1k text prompts from the MSCOCO-2017 validation dataset and evaluate all models on these prompts.

**DiT-based DI++ Model Shows the Best Performance.** As Table 1 shows, the DiT-DI++ model shows a leading HPSv2.0 of **28.48** with only 0.6B parameters, outperforming other open-sourced models which have sizes varying from 0.6B to 10+B. In Table 2, the DiT-DI++ model achieves an ImageReward of **1.24** and an Aesthetic score of **6.19**. Besides, in both Table 1 and 2, the SD1.5-DI++ models show improved scores than SD1.5 diffusion model, Hyper-SD, and SD1.5 diffusion DPO(Wallace et al., 2024).

**The Zero-shot Generalization Ability of Aligned Models.** Another interesting finding of the DI++ algorithm, as well as the human preference alignment of the one-step generator model, is its zero-shot generalization ability. As Table 1 and 2 show, models aligned with Image Reward not only show a dominating Image Reward metric but also show strong aesthetic scores and HPSv2.0. Such a zero-shot generalization ability indicates that if the models are aligned with a sufficiently good reward function with DI++, they can readily generalize well to other human preference scores.

**Comparing with Other Few-step Generator Models.** Table 1 and 2 show the superior advantage of DI++ aligned one-step model over other few-step models. In this paragraph, we give Figure 4 for a qualitative comparison of our models with other few-step models in Table 2. When compared with other few-step generative models, the aligned model shows better aesthetic quality.

**Ablation Comparison** Table 2 shows an ablation comparison of the effects of Diff-Instruct++ with different explicit reward scales $\alpha_r$ and classifier-free guidance reward scales $\alpha_c$. For instance, for the SD1.5 model, we find that a larger explicit reward scale always leads to higher Image Reward and Aesthetic Score, while it may trade off the CLIPScore which represents the image-text semantic alignment. Such an observation indicates a trade-off between the image-text alignment and human-preference alignment if we solely use human-preference reward as the implicit reward. One possible solution is to also add the image-text alignment scores such as the CLIPScore as an explicit reward. Besides, we also find that a CFG reward scale that is too large might hurt the human-preference alignment. For instance, the SD1.5-DI++ one-step model with ($\alpha_r = 100, \alpha_c = 1.5$) shows better score than models with ($\alpha_r = 100, \alpha_c = 4.5$). This indicates that though CFG is practically useful in diffusion sampling, it is not perfectly aligned with human preferences. Therefore, finding a proper CFG reward scale is important for optimal human-preference alignment with Diff-Instruct++.

Table 1: HPS v2 benchmark. We compare open-sourced models regardless of their base model and architecture.† indicates our implementation. Models with bold formats are our models. Numbers with bold formats are the highest scores.

| Model | Animation | Concept-art | Painting | Photo | Average |
|---|---|---|---|---|---|
| GLIDE Nichol et al. (2021) | 23.34 | 23.08 | 23.27 | 24.50 | 23.55 |
| LAFITE Zhou et al. (2022) | 24.63 | 24.38 | 24.43 | 25.81 | 24.81 |
| VQ-Diffusion Gu et al. (2022) | 24.97 | 24.70 | 25.01 | 25.71 | 25.10 |
| FuseDream Liu et al. (2021) | 25.26 | 25.15 | 25.13 | 25.57 | 25.28 |
| Latent Diffusion Rombach et al. (2022) | 25.73 | 25.15 | 25.25 | 26.97 | 25.78 |
| CogView2 Ding et al. (2022) | 26.50 | 26.59 | 26.33 | 26.44 | 26.47 |
| DALL·E mini | 26.10 | 25.56 | 25.56 | 26.12 | 25.83 |
| Versatile Diffusion Xu et al. (2023b) | 26.59 | 26.28 | 26.43 | 27.05 | 26.59 |
| VQGAN + CLIP Esser et al. (2021) | 26.44 | 26.53 | 26.47 | 26.12 | 26.39 |
| DALL·E 2 Ramesh et al. (2022) | 27.34 | 26.54 | 26.68 | 27.24 | 26.95 |
| Stable Diffusion v1.4 Rombach et al. (2022) | 27.26 | 26.61 | 26.66 | 27.27 | 26.95 |
| Stable Diffusion v2.0 Rombach et al. (2022) | 27.48 | 26.89 | 26.86 | 27.46 | 27.17 |
| Epic Diffusion | 27.57 | 26.96 | 27.03 | 27.49 | 27.26 |
| DeepFloyd-XL | 27.64 | 26.83 | 26.86 | 27.75 | 27.27 |
| Openjourney | 27.85 | 27.18 | 27.25 | 27.53 | 27.45 |
| MajicMix Realistic | 27.88 | 27.19 | 27.22 | 27.64 | 27.48 |
| ChilloutMix | 27.92 | 27.29 | 27.32 | 27.61 | 27.54 |
| Deliberate | 28.13 | 27.46 | 27.45 | 27.62 | 27.67 |
| Realistic Vision | 28.22 | 27.53 | 27.56 | 27.75 | 27.77 |
| SDXL-base(Podell et al., 2023) | 28.42 | 27.63 | 27.60 | 27.29 | 27.73 |
| SDXL-Refiner(Podell et al., 2023) | 28.45 | 27.66 | 27.67 | 27.46 | 27.80 |
| Dreamlike Photoreal 2.0 | 28.24 | 27.60 | 27.59 | 27.99 | 27.86 |
| SD15-15Step(Rombach et al., 2022) | 26.76 | 26.37 | 26.41 | 27.12 | 26.66 |
| SD15-25Step(Rombach et al., 2022) | 27.04 | 26.57 | 26.61 | 27.30 | 26.88 |
| SD15-DPO-15Step(Wallace et al., 2024) | 27.11 | 26.75 | 26.70 | 27.30 | 26.97 |
| SD15-DPO-25Step(Wallace et al., 2024) | 27.54 | 26.97 | 26.99 | 27.49 | 27.25 |
| SD15-LCM-1Step(Luo et al., 2023a) | 23.35 | 23.41 | 23.53 | 23.81 | 23.52 |
| SD15-LCM-4Step(Luo et al., 2023a) | 26.42 | 25.79 | 25.95 | 26.91 | 26.27 |
| SD15-TCD-1Step(Zheng et al., 2024) | 23.37 | 23.16 | 23.26 | 23.88 | 23.42 |
| SD15-TCD-4Step(Zheng et al., 2024) | 26.67 | 26.25 | 26.26 | 27.19 | 26.59 |
| SD15-Hyper-1Step(Ren et al., 2024) | 27.76 | 27.36 | 27.41 | 27.63 | 27.54 |
| SD15-Hyper-4Step(Ren et al., 2024) | 28.04 | 27.39 | 27.42 | 27.89 | 27.69 |
| SD15-Instaflow-1Step(Liu et al., 2023) | 26.07 | 25.80 | 25.89 | 26.32 | 26.02 |
| SD15-PeReflow-1Step(Yan et al., 2024) | 25.70 | 25.45 | 25.57 | 25.96 | 25.67 |
| SD15-BOOT-1Step(Gu et al., 2023) | 25.29 | 24.40 | 24.61 | 25.16 | 24.86 |
| SD21-SwiftBrush-1Step(Nguyen & Tran, 2023) | 26.91 | 26.32 | 26.37 | 27.21 | 26.70 |
| SD15-SiDLSG-1Step(Zhou et al., 2024a) | 27.39 | 26.65 | 26.58 | 27.30 | 26.98 |
| SD21-SiDLSG-1Step(Zhou et al., 2024a) | 27.42 | 26.81 | 26.79 | 27.31 | 27.08 |
| SD21-TURBO-1Step(Sauer et al., 2023b) | 27.48 | 26.86 | 27.46 | 26.89 | 27.71 |
| SDXL-DMD2-1Step-1024(Yin et al., 2024) | 27.67 | 27.02 | 27.01 | 26.94 | 27.16 |
| SDXL-DMD2-4Step-1024(Yin et al., 2024) | **28.97** | 27.99 | 27.90 | 28.28 | 28.29 |
| SDXL-DMD2-1Step-512(Yin et al., 2024) | 27.70 | 27.07 | 27.02 | 26.94 | 27.18 |
| SDXL-DMD2-4Step-512(Yin et al., 2024) | 27.22 | 26.65 | 26.62 | 26.57 | 26.76 |
| SD15-DMD2-1Step-512(Yin et al., 2024) | 26.31 | 25.75 | 25.78 | 26.59 | 26.11 |
| PixelArt-$\alpha$-25Step-512(Chen et al., 2023) | 28.77 | 27.92 | 27.96 | 28.37 | 28.25 |
| PixelArt-$\alpha$-15Step-512(Chen et al., 2023) | 28.68 | 27.85 | 27.87 | 28.29 | 28.17 |
| **SD15-DI++-1Step**($\alpha_r = 100, \alpha_c = 1.5$) | 28.42 | 27.84 | 28.01 | 28.19 | 28.12 |
| **DiT-DI++-1Step**($\alpha_r = 10, \alpha_c = 4.5$) | 28.91 | **28.25** | **28.28** | **28.50** | **28.48** |

Table 2: Quantitative comparisons of text-to-image models on 1k MSCOCO-2017 validation prompts. DI++ is short for Diff-Instruct++. $\alpha_r$ and $\alpha_c$ are short for $\alpha_{rew}$ and $\alpha_{cfg}$ in Algorithm 1. † means our implementation. Data means the model needs image data for training. Sampling means the model needs to draw samples from reference diffusion models. Reward means the model needs a human reward model for training.

| Model | Steps | Type | Params | Image Reward | Aes Score | Pick Score | CLIP Score |
|---|---|---|---|---|---|---|---|
| SD15-Base(Rombach et al., 2022) | 15 | UNet | 0.86B | 0.08 | 5.25 | 0.212 | 30.99 |
| SD15-Base(Rombach et al., 2022) | 25 | UNet | 0.86B | 0.22 | 5.32 | 0.216 | 31.13 |
| SD15-DPO(Wallace et al., 2024) | 15 | UNet | 0.86B | 0.20 | 5.29 | 0.214 | 31.07 |
| SD15-DPO(Wallace et al., 2024) | 25 | UNet | 0.86B | 0.28 | 5.37 | 0.218 | 31.25 |
| SD15-LCM(Luo et al., 2023a) | 1 | UNet | 0.86B | -1.58 | 5.04 | 0.194 | 27.20 |
| SD15-LCM(Luo et al., 2023a) | 4 | UNet | 0.86B | -0.23 | 5.40 | 0.214 | 30.11 |
| SD15-TCD(Zheng et al., 2024) | 1 | UNet | 0.86B | -1.49 | 5.10 | 0.196 | 28.30 |
| SD15-TCD(Zheng et al., 2024) | 4 | UNet | 0.86B | -0.04 | 5.28 | 0.212 | 30.43 |
| PeRFlow(Yan et al., 2024) | 4 | UNet | 0.86B | -0.20 | 5.51 | 0.211 | 29.54 |
| SD15-Hyper(Ren et al., 2024) | 1 | UNet | 0.86B | 0.28 | 5.49 | 0.214 | 30.82 |
| SD15-Hyper(Ren et al., 2024) | 4 | UNet | 0.86B | 0.42 | 5.41 | 0.217 | 31.03 |
| SD15-InstaFlow(Liu et al., 2023) | 1 | UNet | 0.86B | -0.16 | 5.03 | 0.207 | 30.68 |
| SD15-SiDLSG(Zhou et al., 2024a) | 1 | UNet | 0.86B | -0.18 | 5.16 | 0.210 | 30.04 |
| SDXL-Base(Rombach et al., 2022) | 25 | UNet | 2.6B | 0.74 | 5.57 | 0.226 | **31.83** |
| SDXL-DMD2-1024(Yin et al., 2024) | 1 | UNet | 2.6B | 0.82 | 5.45 | 0.224 | 31.78 |
| SDXL-DMD2-1024(Yin et al., 2024) | 4 | UNet | 2.6B | 0.87 | 5.52 | **0.231** | 31.50 |
| SDXL-DMD2-512(Yin et al., 2024) | 1 | UNet | 2.6B | 0.36 | 5.03 | 0.215 | 31.54 |
| SDXL-DMD2-512(Yin et al., 2024) | 4 | UNet | 2.6B | -0.18 | 5.17 | 0.206 | 29.28 |
| SD15-DMD2-512(Yin et al., 2024) | 1 | UNet | 2.6B | -0.12 | 5.24 | 0.211 | 30.00 |
| SD21-Turbo(Sauer et al., 2023b) | 1 | UNet | 0.86B | 0.56 | 5.47 | 0.225 | 31.50 |
| PixelArt-$\alpha$-512(Chen et al., 2023) | 25 | DiT | 0.6B | 0.82 | 6.01 | 0.227 | 31.20 |
| PixelArt-$\alpha$-512(Chen et al., 2023) | 15 | DiT | 0.6B | 0.82 | 6.03 | 0.226 | 31.16 |
| **SD15-DI++** ($\alpha_r$=0, $\alpha_c$=1.5) | 1 | UNet | 0.86B | 0.29 | 5.26 | 0.216 | 30.64 |
| **SD15-DI++** ($\alpha_r$=100, $\alpha_c$=1.5) | 1 | UNet | 0.86B | 0.46 | 5.44 | 0.218 | 30.33 |
| **SD15-DI++** ($\alpha_r$=100, $\alpha_c$=4.5) | 1 | UNet | 0.86B | 0.45 | 5.30 | 0.217 | 31.00 |
| **SD15-DI++** ($\alpha_r$=1000, $\alpha_c$=1.5) | 1 | UNet | 0.86B | 0.82 | 5.78 | 0.219 | 30.30 |
| **DiT-DI++** ($\alpha_r$=0, $\alpha_c$=4.5) | 1 | DiT | 0.6B | 0.74 | 5.91 | 0.225 | 31.04 |
| **DiT-DI++** ($\alpha_r$=1, $\alpha_c$=4.5) | 1 | DiT | 0.6B | 0.85 | 6.03 | 0.224 | 30.76 |
| **DiT-DI++** ($\alpha_r$=10, $\alpha_c$=4.5) | 1 | DiT | 0.6B | **1.24** | **6.19** | 0.225 | 30.80 |

## 5.3 Qualitative Evaluations

In this section, we evaluate all one-step text-to-image generator models, showing that the DI++-aligned model shows improved human preference performances. Before the quantitative evaluations, we first give a qualitative comparison of the models with and without alignment.

**The Effects of Human Reward and CFG Reward.** Figure 3 shows a visualization comparison of DiT-based one-step models trained with DI++ with different image reward scales and CFG reward scales. We will give some intuition to better understand the effects of two rewards.

1. From the bottom to the top, the first row of Figure 3* is the generated results from models trained with no image reward and no CFG reward (i.e., scales are set to 0). These images are pretty realistic (such as the coast image), but are of poor details and aesthetic appearance;

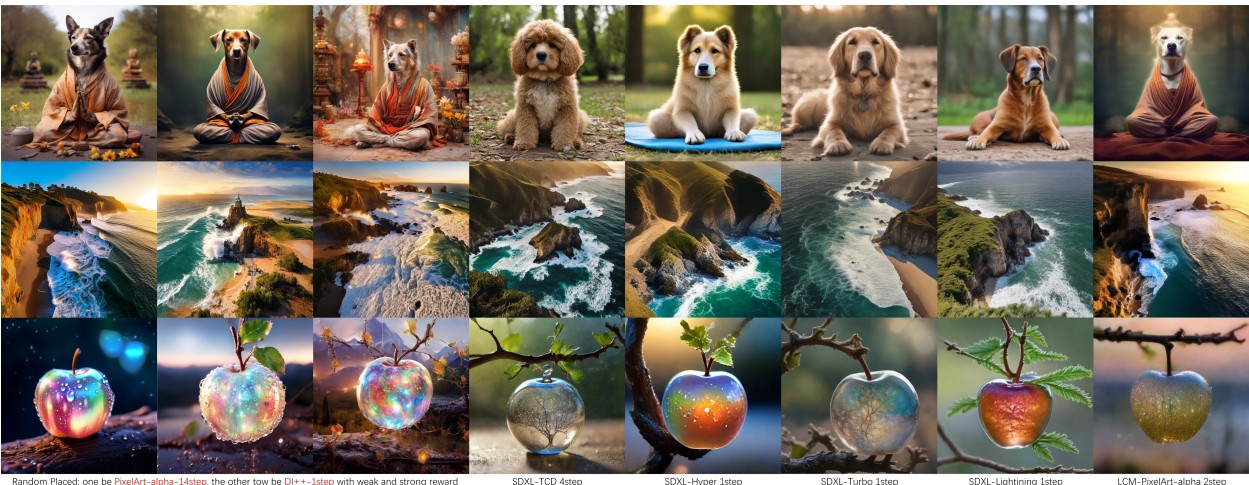

Figure 4: Qualitative comparison of our our Diff-Instruct++ aligned models against other few-step text-to-image models in Table 2. The left three columns are randomly placed, with one generated by PixelArt-$\alpha$ model with 30 steps, one generated by a one-step model aligned with Diff-Instruct++ with a CFG scale of 4.5 and reward scale of 1.0, and another generated by a one-step model aligned with 4.5 CFG and 10.0 reward. Please zoom in to check details, lighting, and aesthetic performances. Could you please tell us which one you like the best? We put the answer for each image and prompts for three rows in Appendix D.4.

2. The second row is the model trained only with a unit scale of image reward. These images are much better than DI++ models with both rewards. However, we can see that solely using image rewards results in vivid colors. This is possible because of the fact that human beings would prefer colorful images with great details. However, only using image rewards results in the objects in the generated images being small, which is possibly not wanted;

3. The third row is the generated images from models trained only with CFG reward. We can see the resulting images are pretty aesthetic. However, these images have too big objects and worse background details;

4. The fourth row is the generated images from models trained with unit image reward together with CFG reward. These images show improved image layout, vivid colors, as well as rich details in both foreground and background. We think these models are of the best results. The quantitative HPS score in Table 1 also confirms our observations;

5. The fifth row is the images from models trained with large image rewards and standard CFG rewards. We find that these images tend to be more like paintings instead of realistic photos. The colors are also much brighter than other models. This shows that a too-large image reward scale may lead to the model generating painting-like images instead of realistic photo-like images. Therefore, researchers are encouraged to carefully choose the image reward scales and the CFG scales to train one-step models with Diff-Instruct++ for their own purposes.

Figure 1 to Figure 6 in the supplementary material shows more comprehensive qualitative results which may bring a deeper understanding of DI++. Due to page limitations, we put more discussions on qualitative findings in Appendix A.4.

### 5.4  Limitations and Failure Cases of Diff-Instruct++

Despite the alignment training stage, the generator model still makes simple mistakes occasionally. Figure 5 shows some bad generation cases that we picked with multiple generation trails. **(1)** The Aligned Model Still Misunderstands the Input Prompt. As the leftmost three images of Figure 5 show, the generated images

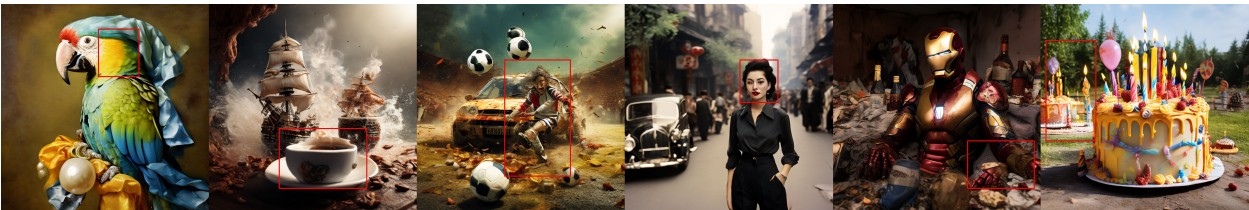

Figure 5: Bad generation cases by aligned one-step generator model (4.5 CFG + 1.0 reward).

ignore the concept of *pear earrings*, the *battling in a coffee cup*, and the *car playing* football. However, we find such a mistake happens only occasionally. **(2)** The Generator Sometimes Generates Bad Human Faces and Hands. Please see the fourth and fifth images of Figure 5. The face of the generated lady in the fourth image is not satisfying with blurred eyes and mouth. In the fifth image, the generated Iron Man character has multiple hands. **(3)** Sometimes The aligned model Still Can not Count Correctly. For instance, in the rightmost image, the prompt asks the model to generate *a birthday cake*, however, the model generates two cakes with one near and another lying far away. Besides, as Figure 3 shows, very strong explicit reward scales lead the image to be very colorful with vivid details. This may cause generated images to look more like paintings instead of realistic photos. Therefore, we suggest researchers carefully choose the explicit reward scales according to their alignment purpose when using Diff-Instruct++.

## 6 Conclusion and Future Works

In this paper, we have presented the Diff-Instruct++ method, the first attempt to align one-step text-to-image generator models with human preference. By formulating the problem as a maximization of expected human reward functions with an IKL divergence regularization, we have developed practical loss functions and a fast-converging yet image data-free alignment algorithm. We also establish theoretical connections of Diff-Instruct++ with previous methods, pointing out that the commonly used classifier-free guidance is secretly doing Diff-Instruct++. Besides, we also introduce a three-stage workflow to develop one-step text-to-image generator models: the pre-training, the reward modeling, and the alignment stage. We train one-step generator models with different alignment configurations and demonstrate the superior advantage of Diff-Instruct++ with a human reward that improves the sample quality and better prompt alignment.

While Diff-Instruct++ does not completely eliminate the occurrence of simple mistakes in image generation, our findings strongly suggest that this approach represents a promising direction for aligning one-step generators with human preferences. We think our work can shed light on future research in improving the responsiveness and accuracy of text-to-image generation models, bringing us closer to AGI systems that can more faithfully interpret and execute human intentions in visual content creation.

**Acknowledgments**

First, we would like to acknowledge Zhengyang Geng for helpful discussions on experiment settings and the overall writing of the paper. We also appreciate the authors of Diff-Instruct(Luo et al., 2023b), Score-implicit Matching(Luo et al., 2024b) and Long-short Score-identity Distillation(Zhou et al., 2024a) for their high-quality codebases on diffusion distillation. Second, we would also like to acknowledge Xiaohongshu Inc. for its support of computing on ablation experiments. Finally, we would like to thank the reviewers and editors of the Diff-Instruct++ paper for their professional opinions and wonderful efforts in organizing the reviewing process.

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

---

**Algorithm 2:** Diff-Instruct++ Pseudo Code.

---

**Input:** prompt dataset $\mathcal{C}$, generator $g_\theta(\boldsymbol{x}_0|\boldsymbol{z}, \boldsymbol{c})$, prior distribution $p_z$, reward model $r(\boldsymbol{x}, \boldsymbol{c})$, reward scale $\alpha_{rew}$, CFG scale $\alpha_{cfg}$, reference diffusion model $\boldsymbol{s}_{ref}(\boldsymbol{x}_t|c, \boldsymbol{c})$, TA diffusion $\boldsymbol{s}_\psi(\boldsymbol{x}_t|t, \boldsymbol{c})$, forward diffusion $p(\boldsymbol{x}_t|\boldsymbol{x}_0)$ (2.1), TA diffusion updates rounds $K_{TA}$, time distribution $\pi(t)$, diffusion model weighting $\lambda(t)$, generator IKL loss weighting $w(t)$.

**while** *not converge* **do**

  fix $\theta$, update $\psi$ for $K_{TA}$ rounds by

  1. sample prompt $\boldsymbol{c} \sim \mathcal{C}$; sample time $t \sim \pi(t)$; sample $\boldsymbol{z} \sim p_z(\boldsymbol{z})$;

  2. generate fake data: $\boldsymbol{x}_0 = \text{sg}[g_\theta(\boldsymbol{z}, \boldsymbol{c})]$; sample noisy data: $\boldsymbol{x}_t \sim p_t(\boldsymbol{x}_t|\boldsymbol{x}_0)$;

  3. update $\psi$ by minimizing loss: $\mathcal{L}(\psi) = \lambda(t)\|\boldsymbol{s}_\psi(\boldsymbol{x}_t|t, \boldsymbol{c}) - \nabla_{\boldsymbol{x}_t} \log p_t(\boldsymbol{x}_t|\boldsymbol{x}_0)\|_2^2$;

  fix $\psi$, update $\theta$ using StaD:

  1. sample prompt $\boldsymbol{c} \sim \mathcal{C}$; sample time $t \sim \pi(t)$; sample $\boldsymbol{z} \sim p_z(\boldsymbol{z})$;

  2. generate fake data: $\boldsymbol{x}_0 = g_\theta(\boldsymbol{z}, \boldsymbol{c})$; sample noisy data: $\boldsymbol{x}_t \sim p_t(\boldsymbol{x}_t|\boldsymbol{x}_0)$;

  3. compute CFG score: $\widetilde{\boldsymbol{s}}_{ref}(\boldsymbol{x}_t|t, \boldsymbol{c}) = \boldsymbol{s}_{ref}(\boldsymbol{x}_t|t, \varnothing) + \alpha_{cfg}\big[\boldsymbol{s}_{ref}(\boldsymbol{x}_t|t, \boldsymbol{c}) - \boldsymbol{s}_{ref}(\boldsymbol{x}_t|t, \varnothing)\big]$;

  4. compute score difference: $\boldsymbol{y}_t := \boldsymbol{s}_\psi(\text{sg}[\boldsymbol{x}_t]|t, \boldsymbol{c}) - \widetilde{\boldsymbol{s}}_{ref}(\text{sg}[\boldsymbol{x}_t]|t, \boldsymbol{c})$;

  5. update $\theta$ by minimizing loss: $\mathcal{L}(\theta) = \big\{ -\alpha_{rew}r(\boldsymbol{x}_0, \boldsymbol{c}) + w(t)\boldsymbol{y}_t^T\boldsymbol{x}_t \big\}$;

**end**
**return** $\theta, \psi$.

---

## Broader Impact Statement

This work is motivated by our aim to increase the positive impact of one-step text-to-image generative models by training them to follow human preferences. By default, one-step generators are either trained over large-scale image-caption pair datasets or distilled from pre-trained diffusion models, which convey only subjective knowledge without human instructions.

Our results indicate that the proposed approach is promising for making one-step generative models more aesthetic, and more preferred by human users. In the longer term, alignment failures could lead to more severe consequences, particularly if these models are deployed in safety-critical situations. For instance, if alignment failures occur, the one-step text-to-image model may generate toxic images with misleading information, and horrible images that can potentially be scary to users. We strongly recommend using our human preference alignment techniques together with AI safety checkers for text-to-image generation to prevent undesirable negative impacts.

## A  Important Materials for Main Content

### A.1  Pseudo-code of Diff-Instruct++

### A.2  Meanings of Hyper-parameters.

**Meanings of Hyper-parameters.**  As we can see in Algorithm 1 (as well as Algorithm 2). Each hyperparameter has its intuitive meaning. The reward scale parameter $\alpha_{rew}$ controls the strength of human preference alignment. The larger the $\alpha_{rew}$ is, the stronger the generator is aligned with human preferences. However, the drawback for a too large $\alpha_{rew}$ might be the loss of diversity and reality. Besides, we empirically find that larger $\alpha_{rew}$ leads to richer generation details and better generation layouts. The CFG scale controls the strength of CFG when computing score functions for the reference diffusion model. As we have shown in Theorem 3.4, the $\alpha_{cfg}$ represents the strength of the implicit reward function (3.5). We empirically

find that the best CFG scale for Diff-Instruct++ is the same as the best CFG scale for sampling from the reference diffusion model. The diffusion model weighting $\lambda(t)$ and the generator loss weighting $w(t)$ controls the strengths put on each time level of updating TA diffusion and the student generator. We empirically find that it is decent to set $\lambda(t)$ to be the same as the default training weighting function for the reference diffusion. And it is decent to set the $w(t) = 1$ for all time levels in practice. In the following section, we give more discussions on Diff-Instruct++.

### A.3   Experiment Details for Pre-training and Alignment

**Experiment Details for Pre-training and Alignment**  We follow the setting of Diff-Instruct (Luo et al., 2024a) to use the same neural network architecture as the reference diffusion model for the one-step generator. The PixelArt-$\alpha$ model is trained using so-called VP diffusion(Song et al., 2020), which first scales the data in the latent space, then adds noise to the scaled latent data. We reformulate the VP diffusion as the form of so-called *data-prediction* proposed in EDM paper (Karras et al., 2022) by re-scaling the noisy data with the inverse of the scale that has been applied to data with VP diffusion. Under the data-prediction formulation, we select a fixed noise $\sigma_{init}$ level to be $\sigma_{init} = 2.5$ following the Diff-Instruct and SiD (Zhou et al., 2024b). For generation, we first generate a Gaussian vector $\boldsymbol{z} \sim p_z = \mathcal{N}(\boldsymbol{0}, \sigma_{init}^2\mathbf{I})$. Then we input $\boldsymbol{z}$ into the generator to generate the latent. The latent vector can then be decoded by the VAE decoder to turn into an image if needed.

**The Training Setup and Costs.**  We train the model with the PyTorch framework. In the pre-training stage, we use the official checkpoint of off-the-shelf PixelArt-$\alpha$-512×512 model [2] as weights of the reference diffusion. We initialize the TA diffusion model with the same weights as the reference diffusion model. We use Diff-Instruct to pre-train the generator. We use the Adam optimizer for both TA diffusion and generation at all stages. For the reference diffusion model, we use a fixed classifier-free guidance scale of 4.5, while for TA diffusion, we do not use classifier-free guidance (i.e., the CFG scale is set to 1.0). We set the Adam optimizer's beta parameters to be $\beta_1 = 0.0$ and $\beta_2 = 0.999$ for both the pre-training and alignment stages. We use a learning rate of $5e-6$ for both TA diffusion and the student one-step generator. For the one-step generator model, we use the adaptive exponential moving average technique by referring to the implementation of the EDM (Karras et al., 2022). We pre-train the one-step model on 4 Nvidia A100 GPUs for two days ($4 \times 48 = 192$ GPU hours), with a batch size of 1024. We find that the Diff-Instruct algorithm converges fast, and after the pre-training stage, the generator can generate images with decent quality.

In the alignment stage, we aim to inspect the one-step generator's behavior with different alignment configurations. Notice that as we have shown in Theorem 3.4, using classifier-free guidance is secretly doing RLHF with Diff-Instruct++, therefore we add both CFG and human reward with different scales to thoroughly study the human preference alignment. More specifically, we align the generator model with five configurations with different CFG scales and reward scales:

1.  no CFG and no reward: use a 1.0 CFG scale and a 0.0 reward scale; This is the weakest setting that we regard the model as a baseline with no human preference alignment;

2.  no CFG and weak reward: use a 1.0 CFG scale and 1.0 reward scale;

3.  strong CFG and no reward: 4.5 CFG scale and 0.0 reward scale;

4.  strong CFG and weak reward: 4.5 CFG scale and 1.0 reward scale;

5.  strong CFG and strong reward: 4.5 CFG scale and 10.0 reward scale.

For all alignment training, we initialize the generator with the same weights that we obtained in the pre-training stage. We put the details of how to construct the one-step generator in Appendix C.1. We also initialize the TA diffusion model with the same weight as the reference diffusion. We use the Image Reward as the human preference reward and use the Diff-Instruct++ algorithm 1 (or equivalently the algorithm 2)

---

[2] https://huggingface.co/PixelArt-alpha/PixelArt-XL-2-512x512

to fine-tune the generator. We also used the Adam optimizer with the parameter $(\beta_1, \beta_2) = (0.0, 0.999)$ for both the generator and the TA diffusion with a batch size of 256. For the alignment stage, we use a fixed exponential moving average decay (EMA) rate of 0.95 for all training trials. After the alignment, the generator aligned with both strong CFG and reward model shows significantly improved aesthetic appearance, better generation layout, and richer image details. Figure 1 shows a demonstration of the generated images using our aligned one-step generator with a CFG scale $\alpha_{cfg}$ of 4.5 and a reward scale $\alpha_{rew}$ of 1.0. We will analyze these models in detail in Section 5.3.

### A.4 More Discussions on Findings of Qualitative Evaluations

There are some other interesting findings when qualitatively evaluate different models. First, we find that the images generated by the aligned model show a better composition when organizing the contents presented in the image. For instance, the main objects of the generated image are smaller and show a more natural layout than other models, with the objects and the background iterating aesthetically. This in turn reveals the human presence: human beings would prefer that the object of an image does not take up all spaces of an image. Second, we find that the aligned model has richer details than the unaligned model. The stronger we align the model, the richer details the model will generate. Sometimes these rich details come as a hint to the readers about the input prompts. Sometimes they just come to improve the aesthetic performance. We think this phenomenon may be caused by the fact that human prefers images with rich details. Another finding is that as the reward scale for alignment becomes stronger, the generated image from the alignment model becomes more colorful and more similar to paintings. Sometimes this leads to a loss of reality to some degree. Therefore, we think that users should choose different aligned one-step models with a trade-off between aesthetic performance and image reality according to the use case.

## B Theory

### B.1 Proof of Theorem 3.1

*Proof.* Recall that $p_\theta(\cdot)$ is induced by the generator $g_\theta(\cdot)$, therefore the sample is obtained by $\boldsymbol{x} = g_\theta(\boldsymbol{z}|\boldsymbol{c}), \boldsymbol{z} \sim p_z$. The term $\boldsymbol{x}$ contains parameter through $\boldsymbol{x} = g_\theta(\boldsymbol{z}|\boldsymbol{c}), \boldsymbol{z} \sim p_z$. To demonstrate the parameter dependence, we use the notation $p_\theta(\cdot)$. $p_{ref}(\cdot)$ is the reference distribution. The alignment objective writes

$$\mathcal{L}(\theta) = \mathbb{E}_{\substack{\boldsymbol{c}, \boldsymbol{z} \sim p_z, \\ \boldsymbol{x} = g_\theta(\boldsymbol{z}|\boldsymbol{c})}} \left\{ r(\boldsymbol{x}, \boldsymbol{c}) + \beta \left[ \log p_\theta(\boldsymbol{x}|\boldsymbol{c}) - \log p_{ref}(\boldsymbol{x}|\boldsymbol{c}) \right] \right\} \tag{B.1}$$

$$= \mathbb{E}_{\boldsymbol{c}, \boldsymbol{z} \sim p_z} \left\{ r(g_\theta(\boldsymbol{z}|\boldsymbol{c}), \boldsymbol{c}) + \beta \left[ \log p_\theta(g_\theta(\boldsymbol{z}|\boldsymbol{c})|\boldsymbol{c}) - \log p_{ref}(g_\theta(\boldsymbol{z}|\boldsymbol{c})|\boldsymbol{c}) \right] \right\} \tag{B.2}$$

Therefore, the $\theta$ gradient of $\mathcal{L}(\theta)$ writes:

$$\frac{\partial}{\partial \theta} \mathcal{L}(\theta) = \frac{\partial}{\partial \theta} \mathbb{E}_{\boldsymbol{c}, \boldsymbol{z} \sim p_z} \left\{ r(g_\theta(\boldsymbol{z}|\boldsymbol{c}), \boldsymbol{c}) + \beta \left[ \log p_\theta(g_\theta(\boldsymbol{z}|\boldsymbol{c})|\boldsymbol{c}) - \log p_{ref}(g_\theta(\boldsymbol{z}|\boldsymbol{c})|\boldsymbol{c}) \right] \right\}$$

$$= \mathbb{E}_{\substack{\boldsymbol{c} \sim p_c, \boldsymbol{z} \sim p_z \\ \boldsymbol{x} = g_\theta(\boldsymbol{z}|\boldsymbol{c})}} \nabla_{\boldsymbol{x}} \left\{ r(\boldsymbol{x}, \boldsymbol{c}) + \beta \left[ \log p_\theta(\boldsymbol{x}|\boldsymbol{c}) - \log p_{ref}(\boldsymbol{x}|\boldsymbol{c}) \right] \right\} \frac{\partial \boldsymbol{x}}{\partial \theta} + \mathbb{E}_{\substack{\boldsymbol{c} \sim p_c, \\ \boldsymbol{x} \sim p_\theta(\cdot|\boldsymbol{c})}} \beta \frac{\partial}{\partial \theta} \log p_\theta(\boldsymbol{x}|\boldsymbol{c}) \tag{B.3}$$

We can see, that the first term of equation (B.3) is the result of Theorem 3.2. Next, we turn to show that the second term of (B.3) will vanish.

$$
\begin{aligned}
\mathbb{E}_{\substack{\boldsymbol{c}\sim p_c \\ \boldsymbol{x}\sim p_\theta(\cdot|\boldsymbol{c})}} \frac{\partial}{\partial\theta}\log p_\theta(\boldsymbol{x}|\boldsymbol{c}) &= \mathbb{E}_{\substack{\boldsymbol{c}\sim p_c \\ \boldsymbol{x}\sim p_\theta(\cdot|\boldsymbol{c})}} \frac{\partial}{\partial\theta}\log p_\theta(\boldsymbol{x}|\boldsymbol{c}) \\
&= \mathbb{E}_{\boldsymbol{c}\sim p_c}\int \frac{1}{p_\theta(\boldsymbol{x}|\boldsymbol{c})}\left\{\frac{\partial}{\partial\theta}p_\theta(\boldsymbol{x}|\boldsymbol{c})\right\}p_\theta(\boldsymbol{x}|\boldsymbol{c})\mathrm{d}\boldsymbol{x} \\
&= \mathbb{E}_{\boldsymbol{c}\sim p_c}\int \left\{\frac{\partial}{\partial\theta}p_\theta(\boldsymbol{x}|\boldsymbol{c})\right\}\mathrm{d}\boldsymbol{x} \qquad\qquad (B.4) \\
&= \mathbb{E}_{\boldsymbol{c}\sim p_c}\frac{\partial}{\partial\theta}\int \left\{p_\theta(\boldsymbol{x}|\boldsymbol{c})\right\}\mathrm{d}\boldsymbol{x} \\
&= \mathbb{E}_{\boldsymbol{c}\sim p_c}\frac{\partial}{\partial\theta}\mathbf{1} \\
&= \mathbf{0} \qquad\qquad\qquad\qquad\qquad\qquad (B.5)
\end{aligned}
$$
$$(B.6)$$

The equality (B.4) holds if function $p_\theta(\boldsymbol{x}|\boldsymbol{c})$ satisfies the conditions (1). $p_\theta(\boldsymbol{x}|\boldsymbol{c})$ is LebeStaue integrable for $\boldsymbol{x}$ with each $\theta$; (2). For almost all $\boldsymbol{x}\in\mathbb{R}^D$, the partial derivative $\partial p_\theta(\boldsymbol{x}|\boldsymbol{c})/\partial\theta$ exists for all $\theta\in\Theta$. (3) there exists an integrable function $h(.):\mathbb{R}^D\to\mathbb{R}$, such that $p_\theta(\boldsymbol{x}|\boldsymbol{c})\le h(\boldsymbol{x})$ for all $\boldsymbol{x}$ in its domain. Then the derivative w.r.t $\theta$ can be exchanged with the integral over $\boldsymbol{x}$, i.e.

$$
\int \frac{\partial}{\partial\theta}p_\theta(\boldsymbol{x}|\boldsymbol{c})\mathrm{d}\boldsymbol{x} = \frac{\partial}{\partial\theta}\int p_\theta(\boldsymbol{x}|\boldsymbol{c})\mathrm{d}\boldsymbol{x}.
$$

$\square$

## B.2 Proof of Theorem 3.2

*Proof.* The proof of Theorem 3.2 is a direct generalization of the proof of Theorem 3.1 as we put in Appendix B.1.

$$
\mathrm{Grad}(\theta) = \mathbb{E}_{\substack{\boldsymbol{c},t,\boldsymbol{z}\sim p_z,\boldsymbol{x}_0=g_\theta(\boldsymbol{z}|\boldsymbol{c}) \\ \boldsymbol{x}_t|\boldsymbol{x}_0\sim p(\boldsymbol{x}_t|\boldsymbol{x}_0)}}\left\{-\nabla_{\boldsymbol{x}_0}r(\boldsymbol{x}_0,\boldsymbol{c})+\beta w(t)\big[\boldsymbol{s}_\theta(\boldsymbol{x}_t|t,\boldsymbol{c})-\boldsymbol{s}_{ref}(\boldsymbol{x}_t|t,\boldsymbol{c})\big]\frac{\partial\boldsymbol{x}_t}{\partial\theta}\right\}
$$

Recall the definition of $p_\theta(\cdot|t,\boldsymbol{c})$, the sample is obtained by $\boldsymbol{x}_0=g_\theta(\boldsymbol{z}|\boldsymbol{c}),\boldsymbol{z}\sim p_z$, and $\boldsymbol{x}_t|\boldsymbol{x}_0\sim p_t(\boldsymbol{x}_t|\boldsymbol{x}_0)$ according to forward SDE (2.1). Since the solution of forward, SDE is uniquely determined by the initial point $\boldsymbol{x}_0$ and a trajectory of Wiener process $\boldsymbol{w}_{t\in[0,T]}$, we slightly abuse the notation and let $\boldsymbol{x}_t=\mathcal{F}(g_\theta(\boldsymbol{z}|\boldsymbol{c}),\boldsymbol{w},t)$ to represent the solution of $\boldsymbol{x}_t$ generated by $\boldsymbol{x}_0$ and $\boldsymbol{w}$. We let $\boldsymbol{w}_{[0,1]}\sim\mathbb{P}_{\boldsymbol{w}}$ to demonstrate a trajectory from the Wiener process where $\mathbb{P}_{\boldsymbol{w}}$ represents the path measure of Weiner process on $t\in[0,T]$. There are two terms that contain the generator's parameter $\theta$. The term $\boldsymbol{x}_t$ contains parameter through $\boldsymbol{x}_0=g_\theta(\boldsymbol{z}|\boldsymbol{c}),\boldsymbol{z}\sim p_z$. The marginal density $p_\theta(\cdot|t,\boldsymbol{c})$ also contains parameter $\theta$ implicitly since $p_\theta(\cdot|t,\boldsymbol{c})$ is initialized with the generator output distribution $p_\theta(\cdot|t=0,\boldsymbol{c})$.

The $p_{ref}(\cdot|t,\boldsymbol{c})$ is defined through the pre-trained diffusion models with score functions $\boldsymbol{s}_{ref}(\cdot|t,\boldsymbol{c})$. The alignment objective between $p_\theta(\cdot|t,\boldsymbol{c})$ and $p_{ref}(\cdot|t,\boldsymbol{c})$ is defined with,

$$
\mathcal{L}(\theta) = \mathbb{E}_{\substack{\boldsymbol{c},t,\boldsymbol{z}\sim p_z,\boldsymbol{x}_0=g_\theta(\boldsymbol{z}|\boldsymbol{c}) \\ \boldsymbol{x}_t|\boldsymbol{x}_0\sim p(\boldsymbol{x}_t|\boldsymbol{x}_0)}}\left\{-r(\boldsymbol{x}_0,\boldsymbol{c})+\beta w(t)\big[\log p_\theta(\boldsymbol{x}_t|t,\boldsymbol{c})-\log p_{ref}(\boldsymbol{x}_t|t,\boldsymbol{c})\big]\right\} \qquad (B.7)
$$

Therefore, the $\theta$ gradient of $\mathcal{L}(\theta)$ writes:

$$\frac{\partial}{\partial\theta}\mathcal{L}(\theta) = \frac{\partial}{\partial\theta}\mathbb{E}_{\substack{\boldsymbol{c},\boldsymbol{z}\sim p_z \\ \boldsymbol{w}\sim\mathbb{P}_{\boldsymbol{w}}}}\left\{r(g_\theta(\boldsymbol{z}|\boldsymbol{c}),\boldsymbol{c}) + \beta\big[\log p_\theta(\mathcal{F}(g_\theta(\boldsymbol{z}|\boldsymbol{c}),\boldsymbol{w},t))|t,\boldsymbol{c}) - \log p_{ref}(\mathcal{F}(g_\theta(\boldsymbol{z}|\boldsymbol{c}),\boldsymbol{w},t))|t,\boldsymbol{c})\big]\right\}$$

$$= \mathbb{E}_{\substack{\boldsymbol{c}\sim p_c,\boldsymbol{z}\sim p_z,\boldsymbol{x}_0=g_\theta(\boldsymbol{z}|\boldsymbol{c}) \\ \boldsymbol{x}_t=\mathcal{F}(\boldsymbol{x}_0,\boldsymbol{w},t))}}\left\{\nabla_{\boldsymbol{x}_0}r(\boldsymbol{x}_0,\boldsymbol{c})\frac{\partial\boldsymbol{x}_0}{\partial\theta} + \beta\big[\nabla_{\boldsymbol{x}_t}\log p_\theta(\boldsymbol{x}_t|\boldsymbol{c}) - \nabla_{\boldsymbol{x}_t}\log p_{ref}(\boldsymbol{x}_t|\boldsymbol{c})\big]\frac{\partial\boldsymbol{x}_t}{\partial\theta}\right\} \quad \text{(B.8)}$$

$$+ \mathbb{E}_{\substack{\boldsymbol{c}\sim p_c,\boldsymbol{z}\sim p_z, \\ \boldsymbol{x}_t\sim p_\theta(\boldsymbol{x}_t|t,\boldsymbol{c})}}\beta\frac{\partial}{\partial\theta}\log p_\theta(\boldsymbol{x}_t|t,\boldsymbol{c}) \quad \text{(B.9)}$$

The first term (B.8) is what we want in the Theorem 3.2. We will show that the second term (B.9) will vanish under mild conditions. The term (B.9) writes

$$\mathbb{E}_{\substack{\boldsymbol{c}\sim p_c,\boldsymbol{z}\sim p_z, \\ \boldsymbol{x}_t\sim p_\theta(\boldsymbol{x}_t|t,\boldsymbol{c})}}\frac{\partial}{\partial\theta}\log p_\theta(\boldsymbol{x}_t|t,\boldsymbol{c}) = \mathbb{E}_{\substack{\boldsymbol{c}\sim p_c \\ \boldsymbol{x}\sim p_\theta(\cdot|\boldsymbol{c})}}\frac{\partial}{\partial\theta}\log p_\theta(\boldsymbol{x}_t|t,\boldsymbol{c})$$

$$= \mathbb{E}_{\boldsymbol{c}\sim p_c}\int\frac{1}{p_\theta(\boldsymbol{x}_t|t,\boldsymbol{c})}\left\{\frac{\partial}{\partial\theta}p_\theta(\boldsymbol{x}_t|t,\boldsymbol{c})\right\}p_\theta(\boldsymbol{x}_t|t,\boldsymbol{c})\mathrm{d}\boldsymbol{x}$$

$$= \mathbb{E}_{\boldsymbol{c}\sim p_c}\int\left\{\frac{\partial}{\partial\theta}p_\theta(\boldsymbol{x}_t|t,\boldsymbol{c})\right\}\mathrm{d}\boldsymbol{x} \quad \text{(B.10)}$$

$$= \mathbb{E}_{\boldsymbol{c}\sim p_c}\frac{\partial}{\partial\theta}\int\left\{p_\theta(\boldsymbol{x}_t|t,\boldsymbol{c})\right\}\mathrm{d}\boldsymbol{x}$$

$$= \mathbb{E}_{\boldsymbol{c}\sim p_c}\frac{\partial}{\partial\theta}\mathbf{1}$$

$$= \mathbf{0} \quad \text{(B.11)}$$

$$\text{(B.12)}$$

The equality (B.10) holds if function $p_\theta(\boldsymbol{x}_t|t,\boldsymbol{c})$ satisfies the conditions (1). $p_\theta(\boldsymbol{x}_t|t,\boldsymbol{c})$ is LebeStaue integrable for $\boldsymbol{x}$ with each $\theta$; (2). For almost all $\boldsymbol{x}_t\in\mathbb{R}^D$, the partial derivative $\partial p_\theta(\boldsymbol{x}_t|t,\boldsymbol{c})/\partial\theta$ exists for all $\theta\in\Theta$. (3) there exists an integrable function $h(.):\mathbb{R}^D\to\mathbb{R}$, such that $p_\theta(\boldsymbol{x}_t|t,\boldsymbol{c})\le h(\boldsymbol{x}_t)$ for all $\boldsymbol{x}_t$ in its domain. Then the derivative w.r.t $\theta$ can be exchanged with the integral over $\boldsymbol{x}_t$, i.e.

$$\int\frac{\partial}{\partial\theta}p_\theta(\boldsymbol{x}_t|t,\boldsymbol{c})\mathrm{d}\boldsymbol{x} = \frac{\partial}{\partial\theta}\int p_\theta(\boldsymbol{x}|\boldsymbol{c})\mathrm{d}\boldsymbol{x}.$$

$\square$

## B.3 Proof of the Pseudo Loss

**Lemma B.1** (Pseudo Loss Function). The pseudo loss function (B.13) has the same $\theta$ gradient as (3.4),

$$\mathcal{L}_p(\theta) = \mathbb{E}_{\substack{\boldsymbol{c},t,\boldsymbol{z}\sim p_z,\boldsymbol{x}_0=g_\theta(\boldsymbol{z}|\boldsymbol{c}) \\ \boldsymbol{x}_t|\boldsymbol{x}_0\sim p(\boldsymbol{x}_t|\boldsymbol{x}_0)}}\left\{-\alpha_{rew}r(\boldsymbol{x}_0,\boldsymbol{c}) + w(t)\boldsymbol{y}_t^T\boldsymbol{x}_t\right\}, \quad \text{(B.13)}$$

$$\widetilde{\boldsymbol{s}}_{ref}(\boldsymbol{x}_t|t,\boldsymbol{c}) = \boldsymbol{s}_{ref}(\boldsymbol{x}_t|t,\varnothing) + \alpha_{cfg}\big[\boldsymbol{s}_{ref}(\boldsymbol{x}_t|t,\boldsymbol{c}) - \boldsymbol{s}_{ref}(\boldsymbol{x}_t|t,\varnothing)\big],$$

$$\boldsymbol{y}_t := \boldsymbol{s}_{\mathrm{sg}[\theta]}(\mathrm{sg}[\boldsymbol{x}_t]|t,\boldsymbol{c}) - \boldsymbol{s}_{ref}(\mathrm{sg}[\boldsymbol{x}_t]|t,\boldsymbol{c}).$$

Here the operator $\mathrm{sg}[\cdot]$ means cutting off all $\theta$ dependence on this variable.

*Proof.* Recall the pseudo loss (B.13):

$$\mathcal{L}_p(\theta) = \mathbb{E}_{\substack{\boldsymbol{c},t,\boldsymbol{z}\sim p_z,\boldsymbol{x}_0=g_\theta(\boldsymbol{z}|\boldsymbol{c}) \\ \boldsymbol{x}_t|\boldsymbol{x}_0\sim p(\boldsymbol{x}_t|\boldsymbol{x}_0)}}\left\{-r(\boldsymbol{x}_0,\boldsymbol{c}) + \beta w(t)\boldsymbol{y}_t^T\boldsymbol{x}_t\right\},$$

$$\boldsymbol{y}_t := \boldsymbol{s}_{\mathrm{sg}[\theta]}(\mathrm{sg}[\boldsymbol{x}_t]|t,\boldsymbol{c}) - \boldsymbol{s}_{ref}(\mathrm{sg}[\boldsymbol{x}_t]|t,\boldsymbol{c})$$

Since all $\theta$ dependence of $\boldsymbol{y}_t$ are cut out, $\boldsymbol{y}_t$ can be regarded as a constant tensor. Taking the $\theta$ gradient of (B.13) leads to

$$
\begin{aligned}
\frac{\partial}{\partial \theta} \mathcal{L}_p(\theta) &= \mathbb{E}_{\substack{\boldsymbol{c},t,\boldsymbol{z}\sim p_z, \boldsymbol{x}_0=g_\theta(\boldsymbol{z}|\boldsymbol{c}) \\ \boldsymbol{x}_t|\boldsymbol{x}_0 \sim p(\boldsymbol{x}_t|\boldsymbol{x}_0)}} \left\{ -\nabla_{\boldsymbol{x}_0} r(\boldsymbol{x}_0, \boldsymbol{c}) \frac{\partial \boldsymbol{x}_0}{\partial \theta} + \beta w(t) \boldsymbol{y}_t^T \frac{\partial \boldsymbol{x}_t}{\partial \theta} \right\} \\
&= \mathbb{E}_{\substack{\boldsymbol{c},t,\boldsymbol{z}\sim p_z, \boldsymbol{x}_0=g_\theta(\boldsymbol{z}|\boldsymbol{c}) \\ \boldsymbol{x}_t|\boldsymbol{x}_0 \sim p(\boldsymbol{x}_t|\boldsymbol{x}_0)}} \left\{ -\nabla_{\boldsymbol{x}_0} r(\boldsymbol{x}_0, \boldsymbol{c}) \frac{\partial \boldsymbol{x}_0}{\partial \theta} + \beta w(t) \Big[ \boldsymbol{s}_{\mathrm{sg}[\theta]}(\mathrm{sg}[\boldsymbol{x}_t]|t, \boldsymbol{c}) - \boldsymbol{s}_{ref}(\mathrm{sg}[\boldsymbol{x}_t]|t, \boldsymbol{c}) \Big]^T \frac{\partial \boldsymbol{x}_t}{\partial \theta} \right\} \\
&= \mathbb{E}_{\substack{\boldsymbol{c},t,\boldsymbol{z}\sim p_z, \boldsymbol{x}_0=g_\theta(\boldsymbol{z}|\boldsymbol{c}) \\ \boldsymbol{x}_t|\boldsymbol{x}_0 \sim p(\boldsymbol{x}_t|\boldsymbol{x}_0)}} \left\{ -\nabla_{\boldsymbol{x}_0} r(\boldsymbol{x}_0, \boldsymbol{c}) \frac{\partial \boldsymbol{x}_0}{\partial \theta} + \beta w(t) \Big[ \boldsymbol{s}_\theta(\boldsymbol{x}_t|t, \boldsymbol{c}) - \boldsymbol{s}_{ref}(\boldsymbol{x}_t|t, \boldsymbol{c}) \Big]^T \frac{\partial \boldsymbol{x}_t}{\partial \theta} \right\}
\end{aligned}
$$

This is the exact gradient term of Diff-Instruct++. $\qquad\square$

### B.4  Proof of Theorem 3.4

*Proof.* Recall the definition of the reward behind classifier-free guidance (3.5). The reward writes

$$
r(\boldsymbol{x}_0, \boldsymbol{c}) = \int_{t=0}^{T} w(t) \log \frac{p_{ref}(\boldsymbol{x}_t|t, \boldsymbol{c})}{p_{ref}(\boldsymbol{x}_t|t)} \mathrm{d}t
$$

This reward will put a higher reward on those samples that have higher class-conditional probability than unconditional probability, therefore encouraging class-conditional sampling. To make the derivation more clear, we consider a single time level $t$, and corresponding

$$
r(\boldsymbol{x}_t, t, \boldsymbol{c}) \coloneqq w(t) \log \frac{p_{ref}(\boldsymbol{x}_t|t, \boldsymbol{c})}{p_{ref}(\boldsymbol{x}_t|t)}. \tag{B.14}
$$

The final result would be an integral of all single time-level $t$. With the single time level, we consider the alignment problem by minimizing

$$
\begin{aligned}
\mathcal{L}(\theta) &= \mathbb{E}_{\boldsymbol{c},\boldsymbol{x}_t \sim p_\theta(\boldsymbol{x}_t|t,\boldsymbol{c})} \big[ -r(\boldsymbol{x}_t, t, \boldsymbol{c}) \big] + \beta w(t) \mathcal{D}_{KL}(p_\theta(\boldsymbol{x}_t|t, \boldsymbol{c}), p_{ref}(\boldsymbol{x}_t|t, \boldsymbol{c})) \\
&= \mathbb{E}_{\boldsymbol{c},\boldsymbol{x}_t \sim p_\theta(\boldsymbol{x}_t|t,\boldsymbol{c})} \big[ -r(\boldsymbol{x}_t, t, \boldsymbol{c}) \big] + \beta w(t) \mathbb{E}_{p_\theta(\boldsymbol{x}_t|t,\boldsymbol{c})} \log \frac{p_\theta(\boldsymbol{x}_t|t, \boldsymbol{c})}{p_{ref}(\boldsymbol{x}_t|t, \boldsymbol{c})}
\end{aligned} \tag{B.15}
$$

The optimal distribution $p_{\theta^*}(\boldsymbol{x}_t|t, \boldsymbol{c})$ that minimize the objective (B.15) will satisfy the equation (B.16)

$$
r(\boldsymbol{x}_t, t, \boldsymbol{c}) = \beta w(t) \log \frac{p_{\theta^*}(\boldsymbol{x}_t|t, \boldsymbol{c})}{p_{ref}(\boldsymbol{x}_t|t, \boldsymbol{c})} + C(\boldsymbol{c}) \tag{B.16}
$$

This means

$$
\log \frac{p_{ref}(\boldsymbol{x}_t|t, \boldsymbol{c})}{p_{ref}(\boldsymbol{x}_t|t)} = \beta \log \frac{p_{\theta^*}(\boldsymbol{x}_t|t, \boldsymbol{c})}{p_{ref}(\boldsymbol{x}_t|t, \boldsymbol{c})} + C(\boldsymbol{c}) \tag{B.17}
$$

The $C(\boldsymbol{c})$ in equation (B.17) is a unknown constant that is independent from $\boldsymbol{x}_t$ and $\theta$. Then we can have the formula for the optimal distribution

$$
\begin{aligned}
\log p_{\theta^*}(\boldsymbol{x}_t|t, \boldsymbol{c}) &= \log p_{\theta^*}(\boldsymbol{x}_t|t, \boldsymbol{c}) + \frac{1}{\beta} \Big\{ \log p_{ref}(\boldsymbol{x}_t|t, \boldsymbol{c}) - \log p_{ref}(\boldsymbol{x}_t|t) \Big\} - \frac{1}{\beta} C(\boldsymbol{c}) \\
&= \log p_{\theta^*}(\boldsymbol{x}_t|t) + (1 + \frac{1}{\beta}) \Big\{ \log p_{ref}(\boldsymbol{x}_t|t, \boldsymbol{c}) - \log p_{ref}(\boldsymbol{x}_t|t) \Big\} - \frac{1}{\beta} C(\boldsymbol{c})
\end{aligned} \tag{B.18}
$$

Besides, we can see that the score function of the optimal distribution writes

$$
\nabla_{\boldsymbol{x}_t} \log p_{\theta^*}(\boldsymbol{x}_t|t, \boldsymbol{c}) = \nabla_{\boldsymbol{x}_t} \log p_{\theta^*}(\boldsymbol{x}_t|t) + (1 + \frac{1}{\beta}) \Big\{ \nabla_{\boldsymbol{x}_t} \log p_{ref}(\boldsymbol{x}_t|t, \boldsymbol{c}) - \nabla_{\boldsymbol{x}_t} \log p_{ref}(\boldsymbol{x}_t|t) \Big\} \tag{B.19}
$$

The equation (B.19) recovers the so-called classifier-free guided score function. The final result is just an integral of the (B.19). Our results show that when using the classifier-free guided score for diffusion distillation using Diff-Instruct (i.e. the equation (3.6)) is secretly doing RLHF (i.e. the Diff-Instruct++) by using the reward (B.14). Besides, our results also bring a new perspective: when sampling from the diffusion model using CFG, the user is secretly doing an inference-time RLHF, and the so-called CFG scale is the RLFH strength.

$\square$

## C    Experiments and Results

### C.1    More Experiment details on Text-to-Image Distillation

We follow the experiment setting of Diff-Instruct (Luo et al., 2024a), generalizing its CIFAR10 experiment to text-to-image generation. Notice that the Diff-Instruct uses the EDM model (Karras et al., 2022) to formulate the diffusion model, as well as the one-step generator. We start with a brief introduction to the EDM model.

The EDM model depends on the diffusion process

$$\mathrm{d}\boldsymbol{x}_t = t\mathrm{d}\boldsymbol{w}_t, t \in [0, T]. \tag{C.1}$$

Samples from the forward process (C.1) can be generated by adding random noise to the output of the generator function, i.e., $\boldsymbol{x}_t = \boldsymbol{x}_0 + t\boldsymbol{\epsilon}$ where $\boldsymbol{\epsilon} \sim \mathcal{N}(\boldsymbol{0}, \boldsymbol{I})$ is a Gaussian vector. The EDM model also reformulates the diffusion model's score matching objective as a denoising regression objective, which writes,

$$\mathcal{L}(\psi) = \int_{t=0}^{T} \lambda(t)\mathbb{E}_{\boldsymbol{x}_0 \sim p_0, \boldsymbol{x}_t|\boldsymbol{x}_0 \sim p_t(\boldsymbol{x}_t|\boldsymbol{x}_0)} \|\boldsymbol{d}_\psi(\boldsymbol{x}_t, t) - \boldsymbol{x}_0\|_2^2 \mathrm{d}t. \tag{C.2}$$

Where $\boldsymbol{d}_\psi(\cdot)$ is a denoiser network that tries to predict the clean sample by taking noisy samples as inputs. Minimizing the loss (C.2) leads to a trained denoiser, which has a simple relation to the marginal score functions as:

$$\boldsymbol{s}_\psi(\boldsymbol{x}_t, t) = \frac{\boldsymbol{d}_\psi(\boldsymbol{x}_t, t) - \boldsymbol{x}_t}{t^2} \tag{C.3}$$

Under such a formulation, we actually have pre-trained denoiser models for experiments. Therefore, we use the EDM notations in later parts.

**Construction of the one-step generator.**    Let $\boldsymbol{d}_\theta(\cdot)$ be pretrained EDM denoiser models. Owing to the denoiser formulation of the EDM model, we construct the generator to have the same architecture as the pre-trained EDM denoiser with a pre-selected index $t^*$, which writes

$$\boldsymbol{x}_0 = g_\theta(\boldsymbol{z}) := \boldsymbol{d}(\boldsymbol{z}, t^*), \quad \boldsymbol{z} \sim \mathcal{N}(\boldsymbol{0}, (t^*)^2 \mathbf{I}). \tag{C.4}$$

We initialize the generator with the same parameter as the teacher EDM denoiser model.

**Time index distribution.**    When training both the EDM diffusion model and the generator, we need to randomly select a time $t$ in order to approximate the integral of the loss function (C.2). The EDM model has a default choice of $t$ distribution as log-normal when training the diffusion (denoiser) model, i.e.

$$t \sim p_{EDM}(t): \quad t = \exp(s) \tag{C.5}$$

$$s \sim \mathcal{N}(P_{mean}, P_{std}^2), \quad P_{mean} = -2.0, P_{std} = 2.0. \tag{C.6}$$

And a weighting function

$$\lambda_{EDM}(t) = \frac{(t^2 + \sigma_{data}^2)}{(t \times \sigma_{data})^2}. \tag{C.7}$$

In our algorithm, we follow the same setting as the EDM model when updating the online diffusion (denoiser) model.

**Weighting function.** For the TA diffusion updates in both pre-training and alignment, we use the same $\lambda_{EDM}(t)$ (C.7) weighting function as EDM when updating the denoiser model. When updating the generator, we use a specially designed weighting function, which writes:

$$w_{Gen}(t) = \frac{1}{\|\boldsymbol{d}_\psi(\text{sg}[\boldsymbol{x}_t], t) - \boldsymbol{d}_{q_t}(\text{sg}[\boldsymbol{x}_t], t)\|_2} \tag{C.8}$$

$$\boldsymbol{x}_t = \boldsymbol{x}_0 + t\epsilon, \quad \epsilon \sim \mathcal{N}(\mathbf{0}, \mathbf{I}) \tag{C.9}$$

The notation sg[·] means stop-gradient of parameter. Such a weighting function helps to stabilize the training.

In the Text-to-Image distillation part, in order to align our experiment with that on CIFAR10, we rewrite the PixelArt-$\alpha$ model in EDM formulation:

$$D_\theta(\mathbf{x}; \sigma) = \mathbf{x} - \sigma F_\theta \tag{C.10}$$

Here, following the iDDPM+DDIM preconditioning in EDM, PixelArt-$\alpha$ is denoted by $F_\theta$, $\mathbf{x}$ is the image data plus noise with a standard deviation of $\sigma$, for the remaining parameters such as $C_1$ and $C_2$, we kept them unchanged to match those defined in EDM. Unlike the original model, we only retained the image channels for the output of this model. Since we employed the preconditioning of iDDPM+DDIM in the EDM, each $\sigma$ value is rounded to the nearest 1000 bins after being passed into the model. For the actual values used in PixelArt-$\alpha$, beta_start is set to 0.0001, and beta_end is set to 0.02. Therefore, according to the formulation of EDM, the range of our noise distribution is [0.01, 156.6155], which will be used to truncate our sampled $\sigma$. For our one-step generator, it is formulated as:

$$g_\theta(\mathbf{x}; \sigma_{\text{init}}) = \mathbf{x} - \sigma_{\text{init}} F_\theta \tag{C.11}$$

Here following Diff-Instruct to use $\sigma_{\text{init}} = 2.5$ and $\mathbf{x} \sim \mathcal{N}(0, \sigma_{\text{init}}\mathbf{I})$, we observed in practice that larger values of $\sigma_{\text{init}}$ lead to faster convergence of the model, but the difference in convergence speed is negligible for the complete model training process and has minimal impact on the final results.

We utilized the SAM-LLaVA-Caption10M dataset, which comprises prompts generated by the LLaVA model on the SAM dataset. These prompts provide detailed descriptions for the images, thereby offering us a challenging set of samples for our distillation experiments.

All experiments in this section were conducted with bfloat16 precision, using the PixelArt-XL-2-512x512 model version, employing the same hyperparameters. For both optimizers, we utilized Adam with a learning rate of 5e-6 and betas=[0, 0.999]. Finally, regarding the training noise distribution, instead of adhering to the original iDDPM schedule, we sample the $\sigma$ from a log-normal distribution with a mean of -2.0 and a standard deviation of 2.0, we use the same noise distribution for both optimization steps and set the two loss weighting to constant 1. Our best model was trained on the SAM Caption dataset for approximately 16k iterations, which is equivalent to less than 2 epochs. This training process took about 2 days on 4 A100-40G GPUs.

With the optimal setting and EDM formulation, we can rewrite our algorithm in an EDM style in Algorithm 3.

## C.2 More Discussions on Experiment Results

**Low Alignment Costs.** Besides the top performance, the training cost with DI++ is surprisingly cheap. Our best model is pre-trained with 4 A100-80G GPUs for 2 days and aligned using the same computation costs. while other industry models in Table 2 require hundreds of A100 GPU days. We summarize the distillation costs in Table 2, marking that DI++ is an efficient yet powerful alignment method with astonishing scaling ability. We believe such efficiency comes from the image-data-free property of DI++. The DI++ does not require image data when aligning, this distinguishes the DI++ from other methods that fine-tune models on highly curated image datasets, which potentially is inefficient.

Table 3: Hyperparameters used for Diff-Instruct++ on Aligning One-step Generator Models

| Hyperparameter | Pre-Training (Diff-Instruct) | | Alignment (Diff-Instruct++) | |
| --- | --- | --- | --- | --- |
| | DM $s_\psi$ | Generator $g_\theta$ | DM $s_\psi$ | Generator $g_\theta$ |
| Learning rate | 5e-6 | 5e-6 | 5e-6 | 5e-6 |
| Batch size | 1024 | 1024 | 256 | 256 |
| $\sigma(t^*)$ | 2.5 | 2.5 | 2.5 | 2.5 |
| Adam $\beta_0$ | 0.0 | 0.0 | 0.0 | 0.0 |
| Adam $\beta_1$ | 0.999 | 0.999 | 0.999 | 0.999 |
| Time Distribution | $p_{EDM}(t)$(C.5) | $p_{EDM}(t)$(C.5) | $p_{EDM}(t)$(C.5) | $p_{EDM}(t)$(C.5) |
| Weighting | $\lambda_{EDM}(t)$(C.7) | 1 | $\lambda_{EDM}(t)$(C.7) | 1 |
| Number of GPUs | 4×A100-40G | 4×A100-40G | 4×H800-80G | 4×H800-80G |

---

**Algorithm 3:** Diff-Instruct++ Pseudo Code under EDM formulation.

---

**Input:** prompt dataset $\mathcal{C}$, generator $g_\theta(\boldsymbol{x}_0|\boldsymbol{z},\boldsymbol{c})$, prior distribution $p_z$, reward model $r(\boldsymbol{x},\boldsymbol{c})$, reward scale $\alpha_{rew}$, CFG scale $\alpha_{cfg}$, reference EDM denoiser model $\boldsymbol{d}_{ref}(\boldsymbol{x}_t|c,\boldsymbol{c})$, TA EDM denoiser $\boldsymbol{d}_\psi(\boldsymbol{x}_t|t,\boldsymbol{c})$, forward diffusion $p(\boldsymbol{x}_t|\boldsymbol{x}_0)$ (2.1), TA EDM denoiser updates rounds $K_{TA}$, time distribution $\pi(t)$, diffusion model weighting $\lambda(t)$, generator IKL loss weighting $w(t)$.

**while** *not converge* **do**

    fix $\theta$, update $\psi$ for $K_{TA}$ rounds by

        1. sample prompt $\boldsymbol{c}\sim\mathcal{C}$; sample time $t\sim\pi(t)$; sample $\boldsymbol{z}\sim p_z(\boldsymbol{z})$;

        2. generate fake data: $\boldsymbol{x}_0 = \text{sg}[g_\theta(\boldsymbol{z},\boldsymbol{c})]$; sample noisy data: $\boldsymbol{x}_t\sim p_t(\boldsymbol{x}_t|\boldsymbol{x}_0)$;

        3. update $\psi$ by minimizing loss: $\mathcal{L}(\psi) = \lambda(t)\|\boldsymbol{d}_\psi(\boldsymbol{x}_t|t,\boldsymbol{c})-\boldsymbol{x}_0\|_2^2$;

    fix $\psi$, update $\theta$ using StaD:

        1. sample prompt $\boldsymbol{c}\sim\mathcal{C}$; sample time $t\sim\pi(t)$; sample $\boldsymbol{z}\sim p_z(\boldsymbol{z})$;

        2. generate fake data: $\boldsymbol{x}_0 = g_\theta(\boldsymbol{z},\boldsymbol{c})$; sample noisy data: $\boldsymbol{x}_t\sim p_t(\boldsymbol{x}_t|\boldsymbol{x}_0)$;

        3. compute CFG score: $\widetilde{\boldsymbol{d}}_{ref}(\boldsymbol{x}_t|t,\boldsymbol{c}) = \boldsymbol{d}_{ref}(\boldsymbol{x}_t|t,\varnothing) + \alpha_{cfg}\big[\boldsymbol{d}_{ref}(\boldsymbol{x}_t|t,\boldsymbol{c}) - \boldsymbol{d}_{ref}(\boldsymbol{x}_t|t,\varnothing)\big]$;

        4. compute score difference: $\boldsymbol{y}_t := \boldsymbol{d}_\psi(\text{sg}[\boldsymbol{x}_t]|t,\boldsymbol{c}) - \widetilde{\boldsymbol{d}}_{ref}(\text{sg}[\boldsymbol{x}_t]|t,\boldsymbol{c})$;

        5. update $\theta$ by minimizing loss: $\mathcal{L}(\theta) = \big\{-\alpha_{rew}r(\boldsymbol{x}_0,\boldsymbol{c}) + w(t)\boldsymbol{y}_t^T\boldsymbol{x}_t\big\}$;

**end**

**return** $\theta,\psi$.

---

## C.3 Pytorch style pseudo-code of Score Implicit Matching

In this section, we give a PyTorch style pseudo-code for algorithm 3.

```
1  import torch
2  import torch.nn as nn
3  import torch.optim as optim
4  import copy
5
6  # Initialize generator G
7  G = Generator()
8
9  ## load teacher DM
10 Drf = DiffusionModel().load('/path_to_ckpt').eval().requires_grad_(False)
11 Dta = copy.deepcopy(Drf) ## initialize online DM with teacher DM
12 r = RewardModel() if alignment_stage else None
```

```
13
14  # Define optimizers
15  opt_G = optim.Adam(G.parameters(), lr=0.001, betas=(0.0, 0.999))
16  opt_Sta = optim.Adam(Dta.parameters(), lr=0.001, betas=(0.0, 0.999))
17
18  # Training loop
19  while True:
20      ## update Dta
21      Dta.train().requires_grad_(True)
22      G.eval().requires_grad_(False)
23
24      ## update TA diffusion
25      prompt = batch['prompt']
26      z = torch.randn((1024, 4, 64, 64), device=G.device)
27      with torch.no_grad():
28          fake_x0 = G(z,prompt)
29
30      sigma = torch.exp(2.0*torch.randn([1,1,1,1], device=fake_x0.device) - 2.0)
31      noise = torch.randn_like(fake_x0)
32      fake_xt = fake_x0 + sigma*noise
33      pred_x0 = Dta(fake_xt, sigma, prompt)
34
35      weight = compute_diffusion_weight(sigma)
36
37      batch_loss = weight * (pred_x0 - fake_x0)**2
38      batch_loss = batch_loss.sum([1,2,3]).mean()
39
40      optimizer_Dta.zero_grad()
41      batch_loss.backward()
42      optimizer_Dta.step()
43
44
45      ## update G
46      Dta.eval().requires_grad_(False)
47      G.train().requires_grad_(True)
48
49      prompt = batch['prompt']
50      z = torch.randn((1024, 4, 64, 64), device=G.device)
51      fake_x0 = G(z, prompt)
52
53      sigma = torch.exp(2.0*torch.randn([1,1,1,1], device=fake_x0.device) - 2.0)
54      noise = torch.randn_like(fake_x0)
55      fake_xt = fake_x0 + sigma*noise
56
57      with torch.no_grad():
58          if use_cfg:
59              pred_x0_rf = Drf(fake_xt, sigma, None) + cfg_scale * (Drf(fake_xt, sigma, prompt
      ) - Drf(fake_xt, sigma, None))
60          else:
61              pred_x0_rf = Drf(fake_xt, sigma, prompt)
62
63          pred_x0_ta = Dta(fake_xt, sigma, prompt)
64
65      denoise_diff = pred_x0_ta - pred_x0_rf
66      weight = compute_G_weight(sigma, denoise_diff)
67
68      batch_loss = weight * denoise_diff * fake_xt
69
70      ## compute reward loss if needed
71      if alignment_stage:
72          reward_loss = -reward_scale * r(fake_x0, prompt)
73          batch_loss += reward_loss
74
75      batch_loss = batch_loss.sum([1,2,3]).mean()
76
77      optimizer_G.zero_grad()
78      batch_loss.backward()
```

```
79    optimizer_G.step()
```

Listing 1: Pytorch Style Pseudo-code of Diff-Instruct++

## D   Prompts

### D.1   Prompts for Figure 1

The prompts are listed from the first row to the second row; from left to right.

- *A small cactus with a happy face in the Sahara desert.*

- *A dog that has been meditating all the time.*

- *A alpaca made of colorful building blocks, cyberpunk.*

- *A dog is reading a thick book.*

- *A delicate apple(universe of stars inside the apple) made of opal hung on a branch in the early morning light, adorned with glistening dewdrops. in the background beautiful valleys, divine iridescent glowing, opalescent textures, volumetric light, ethereal, sparkling, light inside body, bioluminescence, studio photo, highly detailed, sharp focus, photorealism, photorealism, 8k, best quality, ultra detail, hyper detail, hdr, hyper detail.*

- *Drone view of waves crashing against the rugged cliffs along Big Sur's Garay Point beach. The crashing blue waters create white-tipped waves, while the golden light of the setting sun illuminates the rocky shore. A small island with a lighthouse sits in the distance, and green shrubbery covers the cliff's edge. The steep drop from the road down to the beach is a dramatic feat, with the cliff's edges jutting out over the sea. This is a view that captures the raw beauty of the coast and the rugged landscape of the Pacific Coast Highway.*

- *Image of a jade green and gold coloured Fabergé egg, 16k resolution, highly detailed, product photography, trending on artstation, sharp focus, studio photo, intricate details, fairly dark background, perfect lighting, perfect composition, sharp features, Miki Asai Macro photography, close-up, hyper detailed, trending on artstation, sharp focus, studio photo, intricate details, highly detailed, by greg rutkowski.*

- *Astronaut in a jungle, cold color palette, muted colors, detailed, 8k.*

### D.2   Prompts of Figure 3

Prompts of Figure 3, from left to right:

- *A small cactus with a happy face in the Sahara desert*;

- *A delicate apple(universe of stars inside the apple) made of opal hung on a branch in the early morning light, adorned with glistening dewdrops. in the background beautiful valleys, divine iridescent glowing, opalescent textures, volumetric light, ethereal, sparkling, light inside body, bioluminescence, studio photo, highly detailed, sharp focus, photorealism, photorealism, 8k, best quality, ultra detail, hyper detail, hdr, hyper detail*;

- *Drone view of waves crashing against the rugged cliffs along Big Sur's Garay Point beach. The crashing blue waters create white-tipped waves, while the golden light of the setting sun illuminates the rocky shore. A small island with a lighthouse sits in the distance, and green shrubbery covers the cliff's edge. The steep drop from the road down to the beach is a dramatic feat, with the cliff's edges jutting out over the sea. This is a view that captures the raw beauty of the coast and the rugged landscape of the Pacific Coast Highway*;

- *Astronaut in a jungle, cold color palette, muted colors, detailed, 8k*;

- *A parrot with a pearl earring, Vermeer style*;

### D.3 Prompts for Figure 3

- prompt for first row of Figure 3: *A small cactus with a happy face in the Sahara desert.*

- prompt for second row of Figure 3: *An image of a jade green and gold coloured Fabergé egg, 16k resolution, highly detailed, product photography, trending on artstation, sharp focus, studio photo, intricate details, fairly dark background, perfect lighting, perfect composition, sharp features, Miki Asai Macro photography, close-up, hyper detailed, trending on artstation, sharp focus, studio photo, intricate details, highly detailed, by greg rutkowski.*

- prompt for third row of Figure 3: *Baby playing with toys in the snow.*

### D.4 Prompts for Figure 4

The answer for the left three columns:

- the first row from left to right is the one-step model (4.5 CFG and 1.0 reward); the PixelArt-$\alpha$ diffusion with 30 generation steps; the one-step model (4.5 CFG and 10.0 reward);

- the PixelArt-$\alpha$ diffusion with 30 generation steps; the PixelArt-$\alpha$ diffusion with 30 generation steps; the one-step model (4.5 CFG and 10.0 reward); the first row from left to right is the one-step model (4.5 CFG and 1.0 reward);

- the PixelArt-$\alpha$ diffusion with 30 generation steps; the first row from left to right is the one-step model (4.5 CFG and 1.0 reward); the first row from left to right is the one-step model (4.5 CFG and 10.0 reward);

The prompts from up to down are:

- *A dog that has been meditating all the time*;

- *Drone view of waves crashing against the rugged cliffs along Big Sur's Garay Point beach. The crashing blue waters create white-tipped waves, while the golden light of the setting sun illuminates the rocky shore. A small island with a lighthouse sits in the distance, and green shrubbery covers the cliff's edge. The steep drop from the road down to the beach is a dramatic feat, with the cliff's edges jutting out over the sea. This is a view that captures the raw beauty of the coast and the rugged landscape of the Pacific Coast Highway*;

- *A delicate apple(universe of stars inside the apple) made of opal hung on a branch in the early morning light, adorned with glistening dewdrops. in the background beautiful valleys, divine iridescent glowing, opalescent textures, volumetric light, ethereal, sparkling, light inside the body, bioluminescence, studio photo, highly detailed, sharp focus, photorealism, photorealism, 8k, best quality, ultra detail, hyper detail, hdr, hyper detail.*

