

Figure 1: More qualitative comparison of different models trained with Diff-Instruct++.

# Supplementary Materials for Diff-Instruct++

## Move Generated Samples

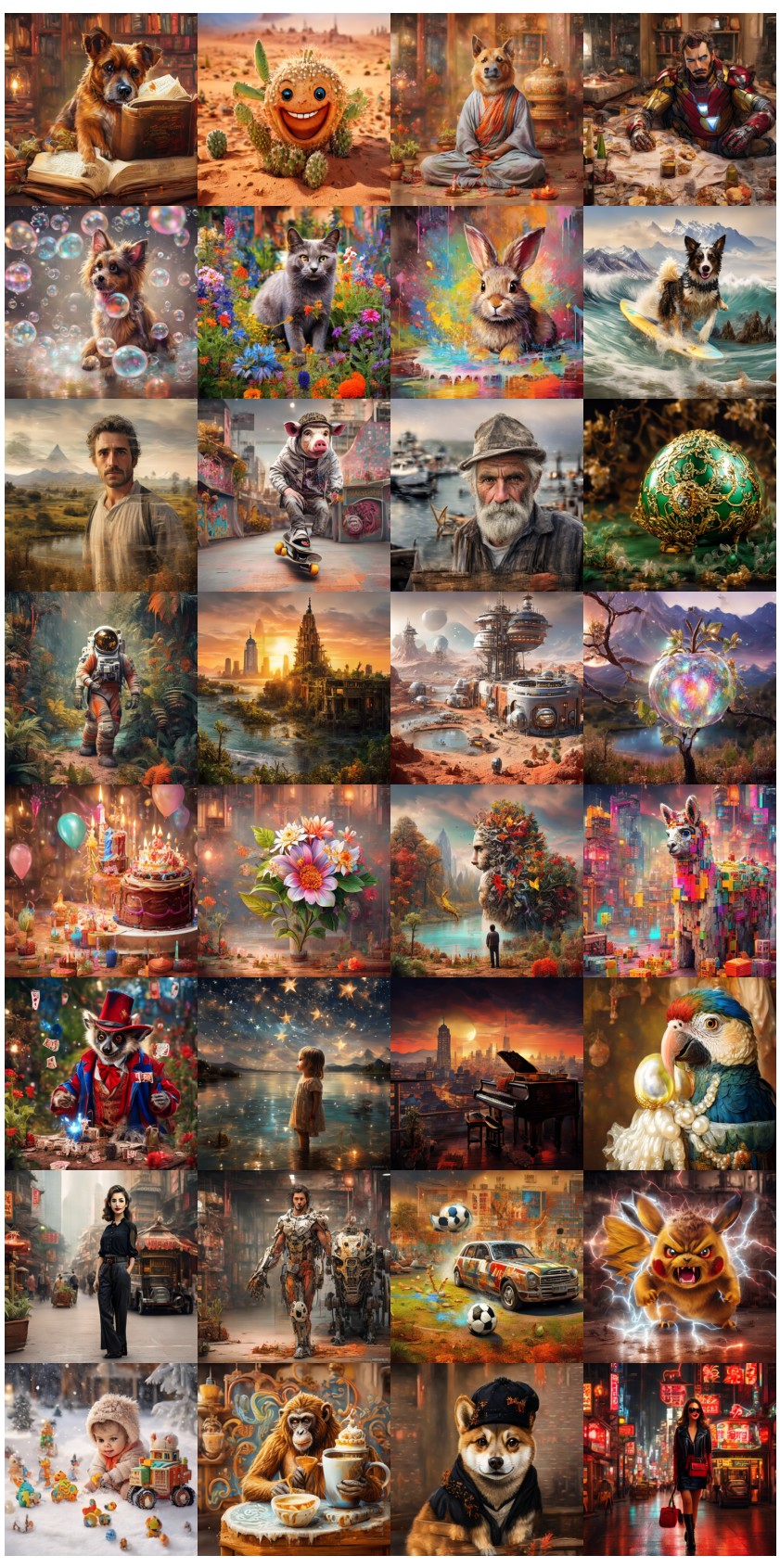

Figure 2: Unpicked images generated by one-step generator model aligned with 4.5 CFG and 10 Reward.

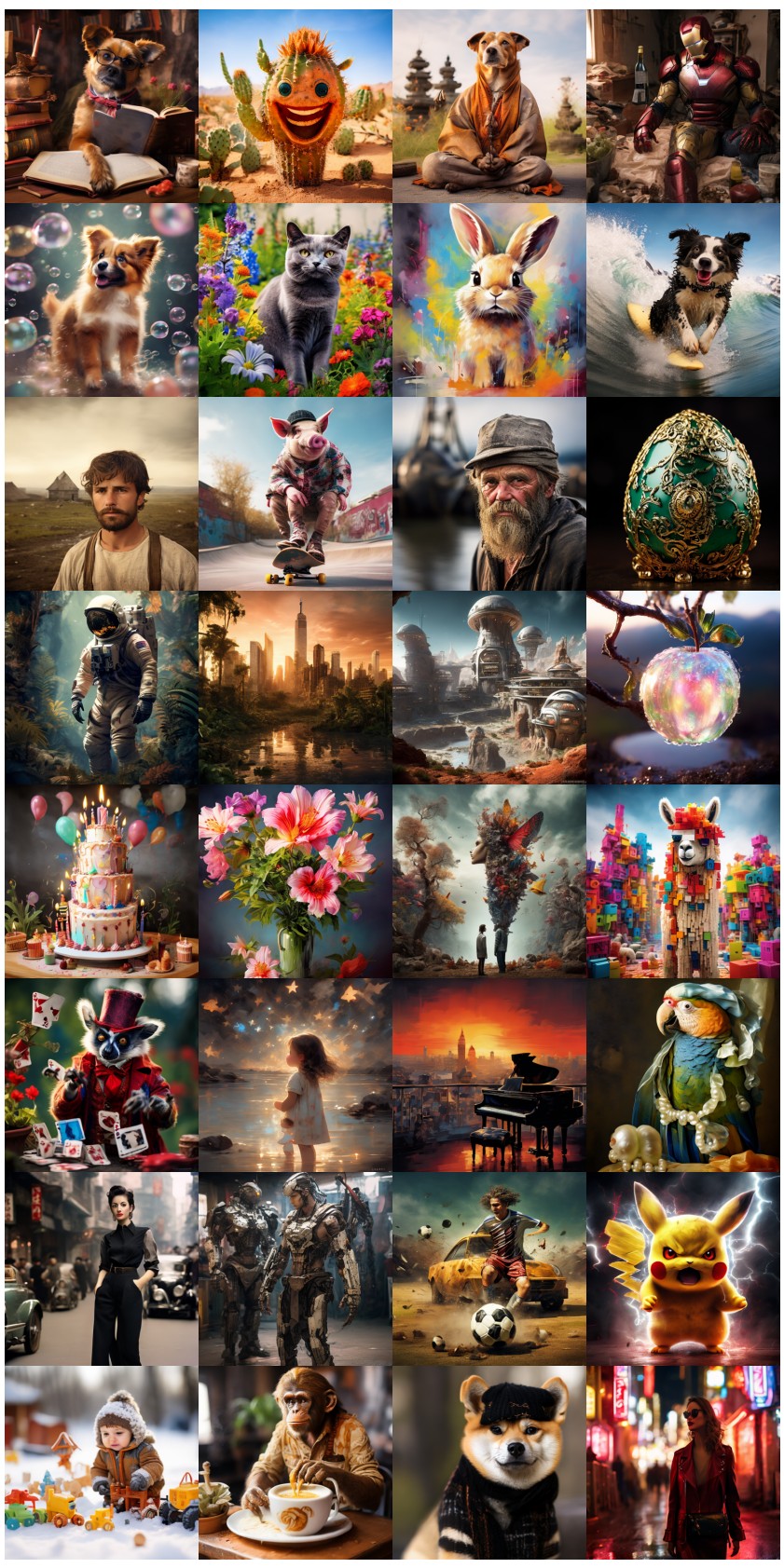

Figure 3: Unpicked images generated by one-step generator model aligned with 4.5 CFG and 1 Reward.

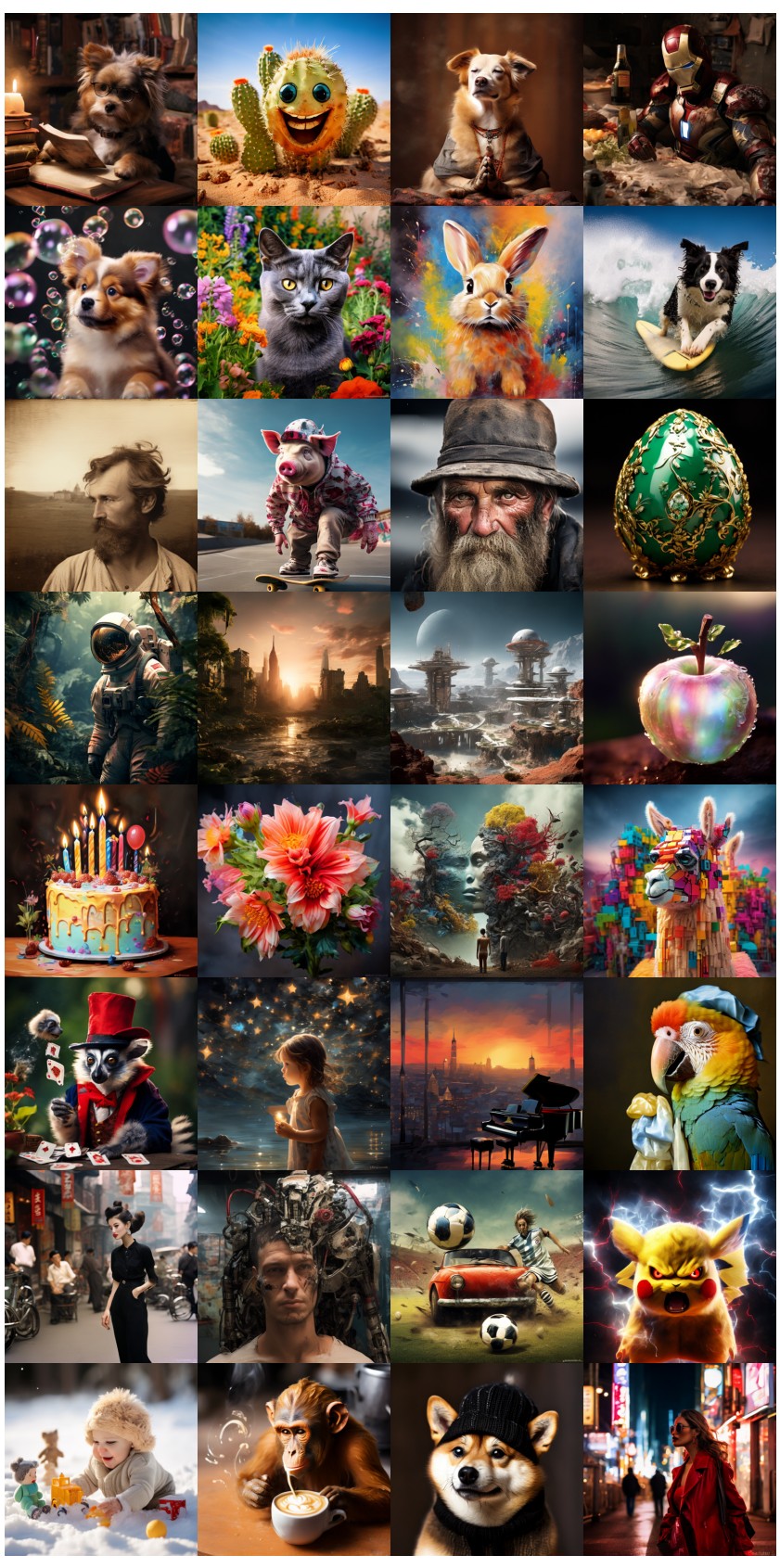

Figure 4: Unpicked images generated by one-step generator model aligned with 4.5 CFG and no Reward.

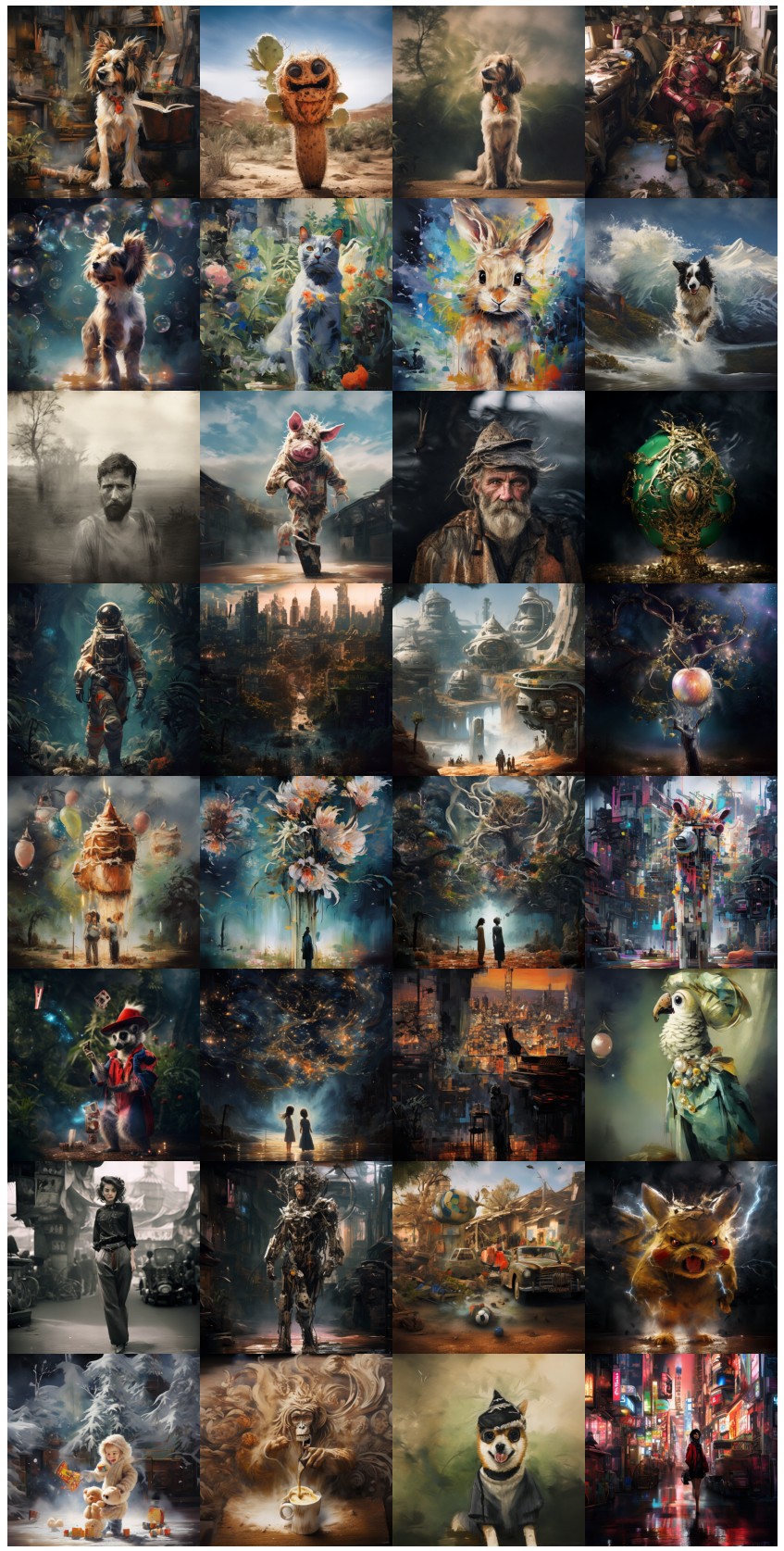

Figure 5: Unpicked images generated by one-step generator model aligned with 1 Reward.

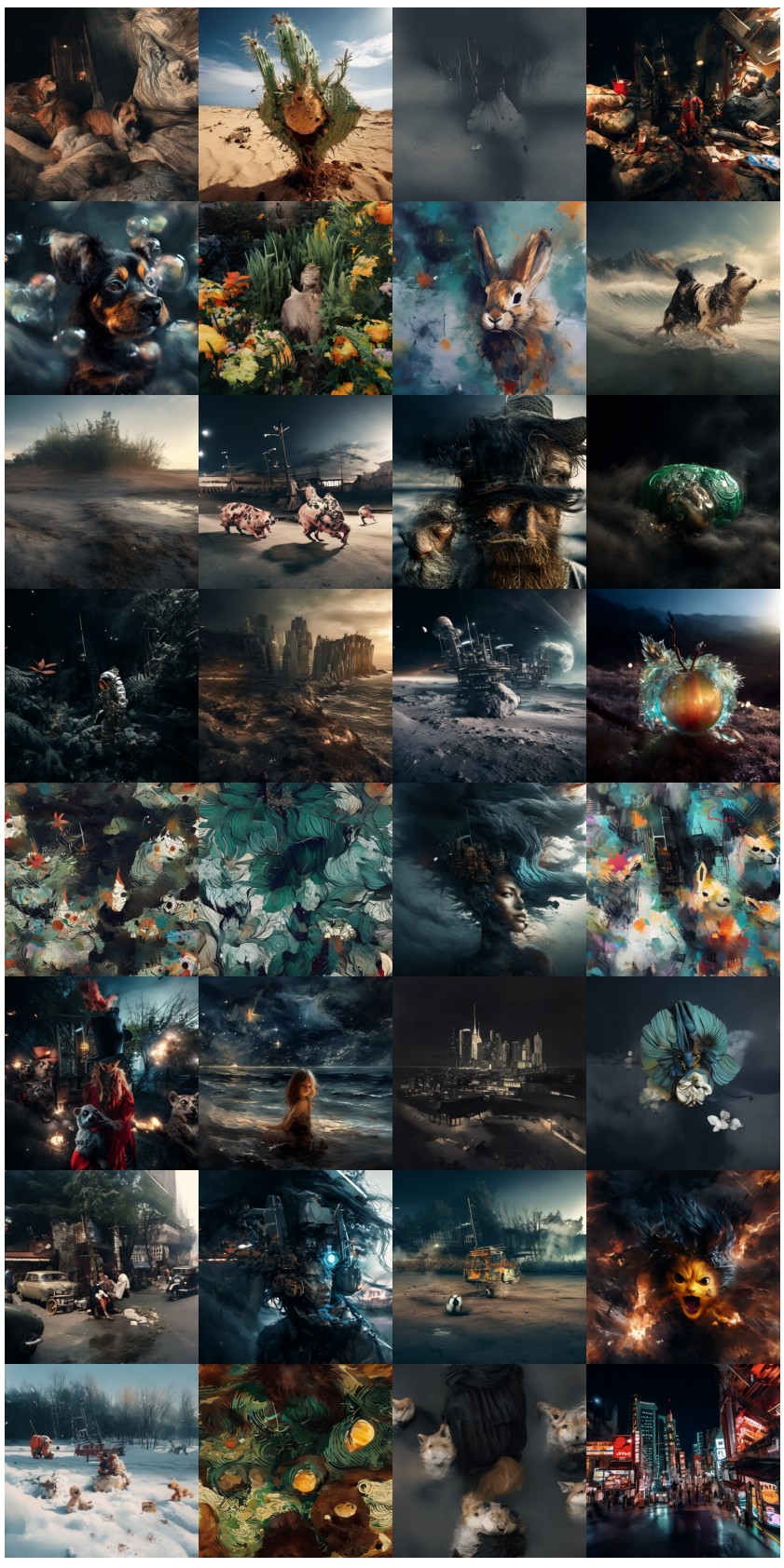

Figure 6: Unpicked images generated by the one-step generator model without alignment with either CFG or reward.