# OpenReview forum: "Diff-Instruct++: Training One-step Text-to-image Generator Model to Align with Human Preferences"
_TMLR — Accepted by TMLR_

### Review · Reviewer_v88H · 2024-08-30

**Summary Of Contributions:**

The paper proposes a method for one-step text-to-image generation that aligns with human preference. Leveraging the knowledge of reinforcement learning using human feedback (RLHF), the authors attempt to maximize the expected human reward functions and introduce a fast-converging and image data-free human preference alignment method for one-step text-to-image generators. Through diffusion distillation and CFG, the experimental results show their superiority results on some human preference metrics compared with other few-step text-to-image generation models.

**Audience:**

Yes

**Broader Impact Concerns:**

If the author can make their code public available, it will be beneficial to the whole community.

**Claims And Evidence:**

Yes

**Requested Changes:**

I highly recommend the authors to solve my 1st and 3rd concerns in the weakness part.

**Strengths And Weaknesses:**

Thanks the authors for the great work on the text-to-image generation model.
Strengths: Besides the technical contributions mentioned above, the paper is also well-polished with adequate theoretical proof. I guess the authors put effort into that.

Weakness:
Though overall a great work, I still think there is some space to improve.
1. Since the Diff-Instruct++ requires a reference diffusion model to initialize, I don't think it's a fair comparison, like in Table 1, in which you compared the DI++ with SDXL-Lightning, SD15-TCD, etc, which initialized from another pre-trained model. Since DI++ is compatible with different reference diffusion models if I understand correctly, the initialized model should be kept the same. For example, if you want to compare DI++ with SD15-TCD, the DI++ should also initialize from SD15.

2. The paper provides some quantitative metrics like Image Reward, Pick Score, etc. However, there are still some other datasets or metrics proposed in these two years for human preference evaluation, like the HPS dataset [1], and the MPS dataset [2] (not sure whether they released their dataset. If not, then forget about it). I would like to see the performance of DI++ on these datasets.

3. The authors mainly compare DI++ with some few-step text-to-image generation methods, which kind of make sense. However, since another target is to generate images aligned with human preference, it's better to compare with some human preference-aligned text-to-image generation methods like [1,2,3,4], although they require multi-steps during generation. If the authors can demonstrate that DI++ can get competitive results with these multi-step human preference-aligned text-to-image generation methods, while the training and inference time is much less, it will strengthen the paper a lot.

[1] Xiaoshi Wu, etal. Better Aligning Text-to-Image Models with Human Preference, ICCV 2023.
[2] Sixian Zhang, etal. Learning Multi-dimensional Human Preference for Text-to-Image Generation, CVPR 2024.
[3] Xiaoshi Wu, etal. Human Preference Score v2: A Solid Benchmark for Evaluating Human Preferences of Text-to-Image Synthesis.
[4] Jiazheng Xu, etal. ImageReward: Learning and Evaluating Human Preferences for Text-to-Image Generation, NIPS 2023.

---

### Review · Reviewer_UNo6 · 2024-09-10

**Summary Of Contributions:**

The authors propose a method to perform RLHF on one step generators. To do so they propose a data free formulation that allow for efficient tuning of a model towards certain rewards. They achieve competitive results aligning Pixart Alpha towards the Image Reward

**Audience:**

Yes

**Claims And Evidence:**

Yes

**Requested Changes:**

HEre are the changes prioritize from most critical to least critical:
- Compare to other RLHF methods
- Compare on standard Image generation metrics
- Improve paper qualitative results
- Multi step diffusion
- Evolution of the different metrics during training (Supp mat maybe)
- Offline

**Strengths And Weaknesses:**

Strength:
- The method seems solid and allows to perform the task on a small budget
- The method allow to perform the reward modeling without the need for training data

Weaknesses:
- The paper lack better qualitative analysis in the main paper. The supp mat figure should be moved to the main paper. I would also like to see failure cases
- The results focus only on the reward scores. However, i think this models should also evaluated with more traditional image generation metrics (FID, CLIPScore, PRDC, GenEval). Indeed, this paper shows that the rewards are maximized but is it at the cost of diversity? Does the model loose clipscore?
- Since the authors use a very high guidance for their best results i suspect a collapse of the model. What about a mix of different guidances during training?
- Also, how does this metrics evolve with training?
- I'm not sure why this method would only work with one step models? If not, would be interested to see results in multi steps setups
- Lack comparaison to other RLHF methods for text to image methods such as Diffusion DPO, and other methods. Indeed this method only compare to none RLHF methods. This method should be compared on Pixart Alpha to have a fair comparaison
- One of the advantages of the method is that it's online. However, if we follow the trends of RLHF in LLMs they tend to be converted to offline methods with training datasets for better efficiency. Would this method work on an offline manner? Or even with a replay buffer for faster iterations?

---

### Review · Reviewer_o47F · 2024-10-07

**Summary Of Contributions:**

The contribution of the paper is
* the development of Diff-Instruct++ (DI++), a method for aligning one-step text-to-image generator models with human preferences.
* DI++ learns human reward functions and aligns the model with an Integral Kullback-Leibler divergence to a reference diffusion model.
* This method enables fast-converging and image data-free human preference alignment for one-step text-to-image generators, leading to significant image generation quality improvements.

**Audience:**

Yes

**Broader Impact Concerns:**

The collection of human preferences may introduce biases. I don't see the paper has consideration of that issue.

**Claims And Evidence:**

Yes

**Requested Changes:**

I think the paper's flow needs to be majorly revised and re-organized. By reading the paper, I don't think the math part has significant contribution, so most of them can be moved to appendix.

The presentation of theorems should be re-considered. And they are not theorems.

The experiment validation needs to be improved. Current experiment section is very hard to understand and hard to tell what's the major benefits of the paper. I suggest more focused evaluation in both qualitative and quantitative results.

I do not see many of the math parts contribute to the significance of the paper, like how to compute the gradients which I believe they are implementation details. So they should be moved to appendix.

**Strengths And Weaknesses:**

Pros
* the approach seems to work
* while the idea of learning a reward is known, the math formulation that leads to an implementable solution is novel and valuable.

Cons
* The idea is straightforward. Learning a reward has long been studied.
* The paper is a bit hard to follow. It seems to me much of the content can be explained in a much easier way, however, the current organization makes it confusing.
* Theorem 3.1 / 3.2 should not be theorems. if they are theorems they should be more self-contained.
* experimental validation is weak. Currently validation is a bit clueless. I think the paper can be improved with more focused evaluation in both qualitative and quantitative.

---

### Decision · Action_Editor_6DTQ · 2024-11-19

**Recommendation:** Accept as is

**Comment:**

While there was limited consensus between reviewers with respect to the breadth of impact of the method, they generally agreed that the paper was correct, and contained enough novel elements and empirical evidence to warrant acceptance. A noted weakness was a lack of investigative experiments ("_why_ does this method work?"), which was counterbalanced by a large number of additional settings and comparative experiments performed during the rebuttal phase.

**Audience:**

It seems likely that this paper will be of interest in the image generation community, and possibly beyond since the proposed techniques could transfer to other domains.

**Claims And Evidence:**

The authors provide pretty extensive empirical support for their main claim, i.e. that the proposed method finetunes one-step generative models given a model of reward, and reaches or surpasses prior art.